# Rehearsal Learning for Avoiding Undesired Future

**Tian Qin, Tian-Zuo Wang, Zhi-Hua Zhou**
National Key Laboratory for Novel Software Technology
Nanjing University, Nanjing, 210023, China
{qint, wangtz, zhouzh}@lamda.nju.edu.cn

## Abstract

Machine learning (ML) models have been widely used to make predictions. Instead of a predictive statement about future outcomes, in many situations we want to pursue a decision: what can we do to avoid the undesired future if an ML model predicts so? In this paper, we present a *rehearsal learning framework*, in which decisions that can persuasively avoid the happening of undesired outcomes can be found and recommended. Based on the *influence* relation, we characterize the generative process of variables with *structural rehearsal models*, consisting of a probabilistic graphical model called *rehearsal graphs* and structural equations, and find actionable decisions that can alter the outcome by reasoning under a Bayesian framework. Moreover, we present a probably approximately correct bound to quantify the associated risk of a decision. Experiments validate the effectiveness of the proposed rehearsal learning framework and the informativeness of the bound.

## 1 Introduction

Machine learning (ML) has achieved great success in various applications, including computer vision [1], natural language processing [2], recommender systems [3, 4], etc. In addition to perception tasks such as classifying an image, ML models have been widely used to make predictions, or forecasting, about future quantities of interest [5, 6]. In many scenarios, however, what we ultimately pursue is typically not attaining merely a predictive statement on what is likely to happen. Instead, as a potential next direction [7], we may seek a way, possibly through decision-making, to avoid the happening of the upcoming undesired future, if the ML model predicts so.

Suppose we use variables $\mathbf{X}$ to predict a future outcome $\mathbf{Y}$ with an ML model $h$. Given an instance $\mathbf{x}$, $h$ outputs a warning signal $\hat{y} = h(\mathbf{x})$ which is outside our desired range. The problem is: what can we do to make the future $\mathbf{Y}$ fall into the desired range? We assume that there is an intermediate stage where one can make actionable decisions to influence $\mathbf{Y}$; *e.g.*, after a sales prediction made at the beginning of a month, there is an intermediate time window before the month-end for the sales manager to take action. We consider decisions in the form of altering variables with fixed values, *e.g.*, setting the discount to 10%. We henceforth call such decisions *alterations*. Let $\mathbf{Z}$ denote the variables in the intermediate stage. The avoiding undesired future (AUF) problem is then how to alter variables in $\mathbf{Z}$ so that the future $\mathbf{Y}$ could be shifted to fall into the desired range.

Common decision-making methods under the umbrella of reinforcement learning (RL) [8] can be applied to the AUF problem but may not be the most appropriate. The reasons include that (a) RL mainly focuses on sequential decision-making tasks while AUF desires a direct decision to change the upcoming future outcome, for which considering decision sequences could be unnecessary or even unrealistic; (b) the success of RL in playing games [9] and autonomous control [10] rely on sufficient interactions between the agent and the environment, but interactions in real AUF problems can be extremely sparse, *e.g.*, the sales manager in the above example can only interact with the environment and make decisions once a month; (c) the Markov decision process (MDP) formalism in RL abstracts decision-making into states, actions, and rewards, which may overlook useful fine-grained structural

37th Conference on Neural Information Processing Systems (NeurIPS 2023).

information in AUF, *e.g.,* the connections between variables that can be altered may help identify useful actions without any exploration. Explicitly incorporating such connections in modeling the AUF problem would be preferable since otherwise approximating them with MDPs may require unnecessarily large state or action spaces.

Therefore, we need a specific method for solving AUF that takes account of the structural information and avoids the use of the large number of interactions. An immediate thought would be seeking cause-effect relations between variables [12], which have been leveraged for some decision-related problems (see Section 5 for a discussion of related work). But as indicated by Zhou [7], causal relations can help but should not be taken as a prerequisite for decision-making for reasons that (a) humans can make good decisions without a thorough or faithful causal understanding of the surrounding environment, sometimes correct decisions can even be made based on an incorrect causal understanding; (b) guiding decisions with causal relations is not always sensible as causation reveals truths that always hold, whereas decision-making is coping with real environments which are often open and dynamic; (c) causal modeling could be helpless for decision-making if the identified causal factors are unactionable; needless to say, it is very difficult to discover true cause-effect relations from data [13]. Recognizing that *correlation* is helpful for prediction but insufficient for decision-making and that *causation* is needed for scientific discovery but too luxury to be relied on for decision-making, Zhou [7] claimed that what is required for decision-making is a kind of intermediate relation that is stronger than correlation but less demanding than causation; this relation was later called *influence relation* [11]. Moreover, Zhou [7] attributed a decision to a series of hypothesized "rehearsal" of possible actions, which is an intuitive way of leveraging or even discovering the influence among variables involved in a decision task. Fig. 1 depicts an intuitive demonstration of the relationship among correlation, influence, and causation.

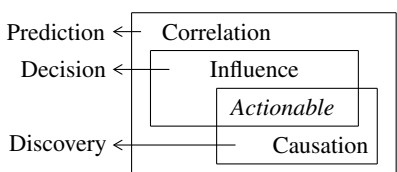

Figure 1: Relationship between correlation, influence, and causation [7, 11].

Based on the concept of rehearsal, we present the structural rehearsal model (SRM) to model the AUF problem. The SRM consists of a new probabilistic graphical model called rehearsal graphs and a collection of generative structural equations. In contrast to traditional structural causal models (SCM) [12], the SRM allows dynamic modeling and accommodates the *influence* relation among variables, effectively capturing the interactions between interrelated but not necessarily causally linked variables. Given the true SRM, the effect of alterations can be calculated or estimated by conducting rehearsals. Consequently, the AUF problem can be addressed by searching for optimal alterations that yield the desired outcomes, though the search process could be complex and difficult.

Note that structural models such as SRMs are generally not available in advance, and learning them from limited observational data is difficult as well. In solving the AUF problem, one needs to tackle challenges posed by limited data, the uncertainty of structural models, and the possibly huge search space for finding an optimal alteration. Moreover, a system outputting decisions should provide some kind of guarantee on the probability of successfully avoiding undesired future, especially in high-stake applications such as economic and safety tasks. The AUF problem can also come in an online fashion and induce an exploration-exploitation trade-off: a decision-maker can choose alterations that will reveal more structural information to benefit future decision-making or alterations that will maximize the success probability in the current decision round.

To address the aforementioned challenges and tackle the AUF problem, we introduce a rehearsal learning framework that integrates structural rehearsal models and Bayesian inference. This framework effectively captures the inherent uncertainty in decision-making processes aimed at averting undesired outcomes by unifying structural modeling, alteration finding, and success probability bounding. Our contributions are as follows:

1. Structural Rehearsal Model: We propose SRM, a novel modeling approach distinct from SCM, designed to model the influence relation, which is more essential than causal relations for decision-making problems.
2. Constrained Optimization for AUF: We formulate a constrained optimization problem to solve the AUF problem, striking a balance between averting undesired future scenarios and accurately learning rehearsal models.

3. Rehearsal Learning Framework: We develop a rehearsal learning framework tailored for solving AUF, showing that building decisions based on the influence relation is practical and feasible. We propose specific learning methods for the basic linear case and show that optimal solutions are not attainable in polynomial time in the framework if joint alterations are allowed, assuming $P \neq NP$. This insight underscores the significance of developing approximate solutions over exact ones.

4. Risk Quantification: We introduce a probably approximately correct bound to quantify the associated risk inherent in a given decision. Our experiments provide empirical evidence of the informativeness of this bound.

## 2 Structural Rehearsal Models

Conventionally, SCMs [12] are used to describe causal relations among variables, based on which actions that affect the outcome could be found. However, in some real-world problems, especially those involving decisions, causal relations are not adequate. For example, let P and Q denote the prices of a product in two stores, respectively. If the first store decreases P to attract customers, the second store will decrease Q accordingly because the second store will lose consumers otherwise. From a causal view, it seems that P causes Q. But obviously, Q also causes P by symmetry. So we have a bi-directional "causal" relation, which is invalid in causal modeling. Mistakes will occur if one applies SCMs to this example as only a one-way causal relation is allowed in SCMs. The issue of such interrelated but not necessarily causally linked variables, as well as several other issues such as the dynamic and time-dependent nature of decision-making, which will be discussed later, are taken into account in the following new structural model called the structural rehearsal model (SRM).

An SRM consists of a set of rehearsal graphs and corresponding structural equations. A rehearsal graph qualitatively describes the relations between variables, and the structural equations characterize the generating process of variables in detail. In contrast to static causal modeling, SRM allows dynamic modeling by defining both the graphs and equations over a time index $t$, which accounts for possible evolutions of the environment. We represent an SRM with $\{\langle G_t, \boldsymbol{\theta}_t \rangle\}_t$, where $G_t$ denotes the rehearsal graph at time $t$ and $\boldsymbol{\theta}_t$ parameterizes corresponding structural equations.

Rehearsal graphs allow both directional and bi-directional edges. The directional edges depict the generation ordering of variables and the bi-directional ones indicate interrelated variables that are mutually influenced. Fig. 2a depicts an example. The definition of rehearsal graphs is given below:

**Definition 1** (Mixed graph). Let $G = (\mathbf{V}, \mathbf{E})$ be a graph, where $\mathbf{V}$ denotes the vertices and $\mathbf{E}$ the edges. $G$ is a *mixed graph* if for any distinct vertices $u, v \in \mathbf{V}$, there is at most one edge connecting them, and the edge is either *directional* ($u \rightarrow v$ or $u \leftarrow v$) or *bi-directional* ($u \leftrightarrow v$).

**Definition 2** (Bi-directional clique). A *bi-directional clique* $C = (\mathbf{V}^c, \mathbf{E}^c)$ of a mixed graph $G = (\mathbf{V}, \mathbf{E})$ is a complete subgraph induced by $\mathbf{V}^c \subseteq \mathbf{V}$ such that any edge $e \in \mathbf{E}^c$ is bi-directional. $C$ is *maximal* if adding any other vertex does not induce a bi-directional clique.

**Definition 3** (Rehearsal graph). Let $G = (\mathbf{V}, \mathbf{E})$ be a mixed graph. Let $\{C_i\}_{i=1}^l$ denote all maximal bi-directional cliques of $G$, where $C_i = (\mathbf{V}_i^c, \mathbf{E}_i^c)$. $G$ is a *rehearsal graph* if and only if:

1. $\mathbf{V}_i^c \cap \mathbf{V}_j^c = \emptyset$ for any $i \neq j$.
2. $\forall i \in [l], u \in \mathbf{V} \setminus \mathbf{V}_i^c$, if there is any edge pointing from $u$ to $\mathbf{V}_i^c$, then $\forall v \in \mathbf{V}_i^c, u \rightarrow v$.
3. The directional edges permit a topological ordering for $\{C_i\}_{i=1}^l$.

Each vertex in a rehearsal graph corresponds to a variable. And the variables are generated following the ordering depicted by the directional edges. The variables in a common maximal bi-directional clique, where only bi-directional edges exist, are mutually influenced instead of having a fixed generating order. Given a rehearsal graph $G$, structural equations accompanying the rehearsal graph are defined over cliques $\{(\mathbf{V}_i^c, \mathbf{E}_i^c)\}_{i=1}^l$. Let $\mathbf{PA}_i^G \triangleq \{u \mid \exists v \in \mathbf{V}_i^c, u \rightarrow v \text{ in } G\}$ denote parents of the $i$-th maximal bi-directional clique. Suppose that the unobserved noise variables are Gaussian, then the structural equation describing the generation process of $\mathbf{V}_i^c$ is parameterized by $\{\boldsymbol{\beta}_i, \boldsymbol{\Sigma}_i\}_{i=1}^l \subseteq \boldsymbol{\theta}$:

$$\mathbf{V}_i^c := f_i(\mathbf{PA}_i^G; \boldsymbol{\beta}_i) + \boldsymbol{\varepsilon}_i, \tag{1}$$

where $f_i$ is a multi-valued function parameterized by $\boldsymbol{\beta}_i$ and $\boldsymbol{\varepsilon}_i \sim \mathcal{N}(0, \boldsymbol{\Sigma}_i)$ denotes the noise.

We denote the operation that alters variable $\mathbf{X}$ with fixed value $\mathbf{x}$ by $Rh(\mathbf{X} = x)$, meaning that we perform a **reh**earsal of setting $\mathbf{X}$ to $\mathbf{x}$ through realistic or hypothetical means. When $Rh(\mathbf{X} = \mathbf{x})$ is

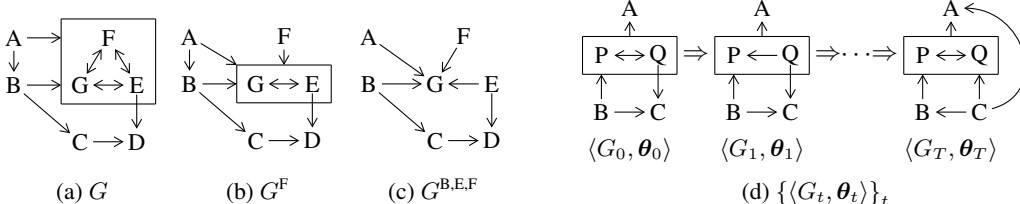

| (a) $G$ | (b) $G^{\mathrm{F}}$ | (c) $G^{\mathrm{B,E,F}}$ | (d) $\{\langle G_t, \boldsymbol{\theta}_t \rangle\}_t$ |
|---|---|---|---|

Figure 2: (a) displays an example rehearsal graph. (b) and (c) displays its alteration graphs with rehearsal operations on $\{\mathrm{F}\}$ and $\{\mathrm{B}, \mathrm{E}, \mathrm{F}\}$ respectively. We use plate notation to simplify the edges: If an endpoint of an edge falls exactly on the boundary of a rectangle, then there are edges for every vertex in that rectangle. (d) shows a full SRM $\{\langle G_t, \boldsymbol{\theta}_t \rangle\}_t$, where $t$ is the time index and the rehearsal graph $G_t$ and parameters $\boldsymbol{\theta}_t$ can evolve over time. Recall the example of two stores and let the variables follow the depicted SRM. Suppose that at $t = 1$, the second store takes a new marketing strategy that no longer follows the price of the first store, then P will not directly affect Q: The generation process evolves to $P \leftarrow Q$ in $G_1$ instead of $P \leftrightarrow Q$ in $G_0$. Then for the first store, a previously correct decision at $t = 0$, which aims to affect C by altering P, is ineffective at $t = 1$.

applied, we remove the incoming arrows of $\mathbf{X}$ in the original rehearsal graph $G$ to get an *alteration rehearsal graph* $G^{\mathbf{X}=\mathbf{x}}$, and the accompanying structural equations are modified according to the newly introduced parental relations in $G^{\mathbf{X}=\mathbf{x}}$, where a new set of parameters in $\boldsymbol{\theta}$ could be introduced for the unseen parental relations to offer more flexible modeling abilities. The distribution $\mathbb{P}(\mathbf{V} \mid Rh(\mathbf{X} = \mathbf{x}))$ of all variables after applying $Rh(\mathbf{X} = \mathbf{x})$ can be derived from Eq. (1). Example illustrations are in Figs. 2b-2c. The "invalid" bi-directional "causal" relations are valid in rehearsal graphs: $P \leftrightarrow Q$ becomes $P \rightarrow Q$ when applying $Rh(P)$ and becomes $P \leftarrow Q$ when applying $Rh(Q)$, capturing the interrelated *influence* between the two variables.

An important feature of SRM that distinguishes it from SCM is the capability of modeling dynamic and time-dependent real-world decision-making environments, where variable relations may evolve and correct decisions can vary dramatically at different times even for the same problem [14]. It is worth noting that these dynamic environments should not be confused with the conventional notion of dynamic systems, where the relationships among variables simply repeat over time. As causal modeling seeks to identify enduring cause-and-effect relationships, SCMs can be helpful in tasks such as "AI for science" [15, 16, 17], where the objective is to uncover persistent scientific truths. However, SCMs can be restrictive in describing the dynamic nature of decision problems. In contrast, the dynamic and time-dependent feature of environments is captured by SRM using evolving rehearsal graphs and corresponding parameters: At time $t$, the generating process characterized by $\langle G_t, \boldsymbol{\theta}_t \rangle$ can differ from previous ones. Fig. 2d illustrates an example where correct decisions vary across time and an SRM $\{\langle G_t, \boldsymbol{\theta}_t \rangle\}_t$ is used to model the dynamic decision environment.

Although $\boldsymbol{\theta}_t$ can be learned from data with general supervised learning methods given a fixed graphical structure, generally rehearsal graphs could not be uniquely identified from data. We present basic properties of rehearsal graphs and a preliminary graph class learning procedure in Appendix A.

## 3 The AUF Problem

We treat AUF as a multi-round online decision-making problem, where in the $t$-th decision round, an agent makes decisions to affect the outcome if it receives an undesired prediction, serving as a warning signal, from an ML model. In real problems, variables involved in each decision round generally do not appear simultaneously; *e.g.*, if a single decision round spans a full month, then a decision-maker may encounter new variables every day. Ignoring the variable generation order can be problematic, but modeling the decision process with refined time granularity is too demanding.

Therefore, we make a midway proposal. We identify two important time points in AUF, the time the ML prediction is made and the time just before the generation of the concerned outcome, so the variables appearing in the $t$-th decision round fall into three consecutive time segments separated by the two time points: $\mathbf{X}_t$, variables appeared before the prediction is made; $\mathbf{Z}_t$, variables appeared after the prediction and before the generation of the outcome; and $\mathbf{Y}_t$, the outcome variable. After the prediction, if the agent decides to do something to change the future outcome, the only variables it can alter are those in $\mathbf{Z}_t$, since $\mathbf{X}_t$ has already happened and cannot be altered. For example, suppose that in the $t$-th month, a sales manager predicts the month sales $\mathbf{Y}_t$ on the first day of that month and designs promotions in that month accordingly. Then $\mathbf{X}_t$ can be marketing variables that appeared

before the first day and $\mathbf{Z}_t$ can be variables in the remaining days of that month, such as the price next week, which can be altered to affect month sales.

More specifically, the decision process in the $t$-th round is: An agent first observes $\mathbf{X}_t$, with which an ML model gives a prediction $\hat{\mathbf{Y}}_t$ for the outcome; if $\hat{\mathbf{Y}}_t \notin \mathcal{S}$, where $\mathcal{S}$ denotes the set of desired values, the agent should perform alterations on $\mathbf{Z}_t$ to prevent $\mathbf{Y}_t \notin \mathcal{S}$; after the alteration, the agent can observe full $\mathbf{Z}_t$ and the outcome $\mathbf{Y}_t$. We assume that the generation mechanism of the encountered variables can be described by an SRM $\{\langle G_t, \boldsymbol{\theta}_t \rangle\}_t$, where $\langle G_t, \boldsymbol{\theta}_t \rangle$ characterizes the generation process in the $t$-th round. Fig. 3 depicts an example. For simplicity, we assume that the environment is stable, *i.e.*, $G_t = G$,

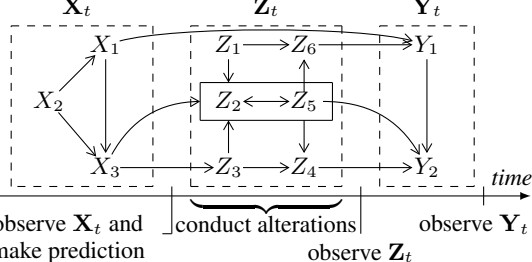

Figure 3: Rehearsal graph $G_t$ in the $t$-th round. The timeline shows the order in which the three sets of variables are generated and events that happened to an agent at different times in *one round*.

$\boldsymbol{\theta}_t = \boldsymbol{\theta}$ for all $t$. Note that the agent does not have access to the true SRM, instead, it can access historical data $D = \{(\mathbf{x}^i, \mathbf{z}^i, \mathbf{y}^i)\}_{i=1}^m$, from which some information about the SRM can be learned.

In real scenarios, some variables cannot be manually altered or cannot be set to certain values. We denote the set of feasible alteration values for an actionable variable $Z_i \in \mathbf{Z}$ by $\Delta(Z_i)$, which we assume to be a closed interval. An alteration is denoted by $\xi_t = \{(Z_{a_i}, z_{a_i})\}_{i=1}^k$, where $Z_{a_i} \in \mathbf{Z}$ is the variable to be altered, $z_{a_i} \in \Delta(Z_{a_i})$ is the alteration value, and $k$ is the alteration size. Applying $\xi_t$ is treated as performing a rehearsal $Rh(\xi_t) \triangleq Rh(\{Z_{a_i} = z_{a_i}\}_{i=1}^k)$ on $G$.

The overall AUF problem is then given $D$ and $\mathbf{x}_t$, $t = 1, \ldots, T$, that arrives sequentially, find alterations $\xi_t$ in each round $t$ to successfully avoid the undesired future as many times as possible. The problem can be formulated as

$$\max_{\xi_{1:T}} \quad \mathbb{E}_{\boldsymbol{\varepsilon}_{1:T}} \left[ \sum_{t=1}^{T} \mathbb{I} \left( \mathbf{Y}_t \in \mathcal{S} \mid \mathbf{x}_t, Rh(\xi_t) \right) \right], \tag{2}$$

where $\boldsymbol{\varepsilon}$ is the noise in Eq. (1) and $\mathbb{I}(\cdot)$ is the indicator function.

If the underlying SRM is known, one can obtain $\mathbb{P}(\mathbf{Y}_t \mid \mathbf{x}_t, Rh(\xi_t))$, and the only issue is searching for a $\xi_t$ that maximizes $\mathbb{P}(\mathbf{Y}_t \in \mathcal{S} \mid \mathbf{x}_t, Rh(\xi_t))$. The main obstacle to solving AUF, however, is the uncertainty of the SRM. The uncertainty leads to an exploration-exploitation trade-off: In each round, if the agent chooses alterations that help reveal true SRMs (exploration), a more accurate environment model may be obtained and benefit decisions in future rounds, but it may fail in the current round. On the other hand, if the agent chooses a short-sighted decision that successfully avoids the upcoming undesired future (exploitation), it may learn little about SRMs, making future optimizations difficult. In addition, the effect of alterations can propagate to downstream variables through the directional edges in a rehearsal graph, thus the value of unactionable variables may be affected by altering the actionable ones. A feasible approach should take account of the uncertainty, exploration-exploitation trade-off, and propagation of effects in the online decision process and yield reasonable decisions.

Some extra knowledge can be utilized. First, the partition of the three sets of variables reflects the order in which the variables are generated. Any edge crossing the three sets should point from the previous one to the latter one, which reduces the uncertainty on $G$. Further, since an agent observes $\mathbf{X}_t$ before making a decision, the generating mechanism inside $\mathbf{X}_t$ does not affect the distribution of outcomes, as shown in Prop. 4, so we can safely pretend that $G$ does not have edges inside $\mathbf{X}_t$.

**Proposition 4.** *For any $\langle G_1, \boldsymbol{\theta}_1 \rangle$ and $\langle G_2, \boldsymbol{\theta}_2 \rangle$ on $\mathbf{X}_t \cup \mathbf{Z}_t \cup \mathbf{Y}_t$, if they differ only in describing the generation of $\mathbf{X}_t$, then for any $\mathbf{X}_t = \mathbf{x}_t$ and alterations $\xi$ on $\mathbf{Z}_t$, we have*

$$\mathbb{P}\left(\mathbf{y}_t \mid \mathbf{x}_t, Rh(\xi); G_1, \boldsymbol{\theta}_1\right) = \mathbb{P}\left(\mathbf{y}_t \mid \mathbf{x}_t, Rh(\xi); G_2, \boldsymbol{\theta}_2\right).$$

In the following, we restrict our attention to a basic instance of the AUF problem: The structural equations $f_i$s are linear and the desired set $\mathcal{S}$ is a convex polytope, *i.e.*,

$$\mathbf{V}_i^c := \boldsymbol{\beta}_i^T \mathbf{PA}_i^G + \varepsilon_i, \ \mathcal{S} = \{\mathbf{y} \in \mathbb{R}^{|\mathbf{Y}|} \mid \mathbf{My} \leq \mathbf{d}\}, \tag{3}$$

where $|\cdot|$ denotes the cardinality of a set, $\mathbf{d} \in \mathbb{R}^s$, $\mathbf{M} \in \mathbb{R}^{s \times |\mathbf{Y}|}$, and $\boldsymbol{\beta}_i \in \mathbb{R}^{|\mathbf{PA}_i^G| \times |\mathbf{V}_i^c|}$.

## 4 Rehearsal Learning

In this section, we present a rehearsal learning framework to address the AUF problem in Eq. (2) and present specific methods for the linear case in Eq. (3), in which we use sampling and rehearsal of actions as a basic building block to find and evaluate decisions.

To account for the uncertainty introduced by limited data, we resort to Bayesian inference. The state of knowledge of the SRM is described by the posterior distribution:

$$\mathbb{P}(G, \boldsymbol{\theta} \mid D_t) = \mathbb{P}(G \mid D_t)\mathbb{P}(\boldsymbol{\theta} \mid G, D_t), \tag{4}$$

where $D_t$ is the evidence collected until the beginning of round $t$, consisting of the initial observational data $D$ and $\{(\mathbf{x}_i, \xi_i, \mathbf{z}_i, \mathbf{y}_i)\}_{i=1}^{t-1}$. To handle the exploration-exploitation trade-off between choosing alterations that help recovers the true SRM and alterations that can avoid undesired future in the current round, we leave the choice to the agent by introducing a hyper-parameter $\tau$ that balances these two kinds of alterations. In the $t$-th round, we propose to solve

$$\max_{\xi_t} \quad \mathrm{I}\left(\mathbf{G}, \boldsymbol{\Theta}; \mathbf{Z}_t, \mathbf{Y}_t \mid D_t, \mathbf{x}_t, Rh(\xi_t)\right) \tag{5}$$

$$\text{s.t.} \quad \mathbb{E}_{G, \boldsymbol{\theta} \mid D_t}\left[\mathbb{I}\left(\mathbf{Y}_t \in \mathcal{S} \mid G, \boldsymbol{\theta}, \mathbf{x}_t, Rh(\xi_t)\right)\right] \geq \tau. \tag{6}$$

The constraint in Eq. (6) reflects the required performance: Performing the alteration should make the belief that the desired future can be successfully achieved greater than $\tau$ given knowledge of the SRM so far. The objective function in Eq. (5) denotes the mutual information between the SRM and the observations revealed after the alteration. Overall, the solution to the optimization problem should avoid undesired future and meanwhile be most informative about the true SRM.

The proposed optimization problem introduces computational challenges, *e.g.*, one has to sum over a super-exponential number of rehearsal graphs to compute the mutual information or the success probability. We divide rehearsal learning into three approximately tractable components: Bayesian update of SRMs, candidate alteration selection, and mutual information maximization. To better guide decision-making, we also present a sampling-based probably approximately correct (PAC) bound to quantify the risk associated with the selected alteration.

### 4.1 Bayesian Update of SRMs

Recall that the posterior of SRMs is divided into the posterior of graphs $\mathbb{P}(G \mid D_t)$ and that of structural equation parameters $\mathbb{P}(\boldsymbol{\theta} \mid G, D_t)$ in Eq. (4). For the former term, we adopt a bootstrapping-based approximation [18], which learns multiple rehearsal graphs on bootstrapped data to build a set of graphs $\mathcal{G}$ and weigh each graph $G \in \mathcal{G}$ equally. The approximated posterior of graphs is

$$\mathbb{P}(G \mid D_t) \approx \frac{1}{|\mathcal{G}|} \sum_{G' \in \mathcal{G}} \mathbb{I}(G' = G). \tag{7}$$

Consequently, for the second term $\mathbb{P}(\boldsymbol{\Theta} \mid \mathbf{G}, D_t)$, we only need to estimate parameters corresponding to graphs that appear in $\mathcal{G}$, which greatly reduces computation. The parameter posterior factorizes as

$$\mathbb{P}(\boldsymbol{\theta} \mid G, D_t) = \prod_{G' \in \mathcal{A}(G), \mathbf{V}^c \in G'} \mathbb{P}(\boldsymbol{\beta}_{\mathbf{V}^c}, \boldsymbol{\Sigma}_{\mathbf{V}^c} \mid G', D_t),$$

where $\mathcal{A}(G)$ is a set of graphs that includes $G$ and related alteration graphs and $\mathbf{V}^c$ denotes a maximal bi-directional clique. $\mathbb{P}(\boldsymbol{\beta}_{\mathbf{V}^c}, \boldsymbol{\Sigma}_{\mathbf{V}^c} \mid G', D_t)$ is the posterior of a set of regression parameters and noise variance given data of $\mathbf{V}^c$ and its parents in $G'$. So various Bayesian learning methods [19] are applicable for its estimation. We adopt Bayesian ridge regression [20] to learn the parameters. In addition, as graph learning is time-consuming, we use an incremental implementation: We build an initial graph posterior from purely observational data for $t = 0$ using Eq. (7), then update the posterior with

$$\mathbb{P}(G \mid D_t) \propto \mathbb{P}(G \mid D_{t-1}) \cdot \frac{1}{n} \sum_{i=1}^{n} \mathbb{P}(\mathbf{z}_t, \mathbf{y}_t \mid \mathbf{x}_t, \xi_t, G, \boldsymbol{\theta}^i),$$

where $\boldsymbol{\theta}^i$ is sampled from $\mathbb{P}(\boldsymbol{\theta} \mid G, D_{t-1})$ and the averaged summation term is an empirical estimate of likelihood. With the above approximations and learning, we effectively obtain an approximate posterior and an efficient sampler over SRMs, which forms the basis of the following two steps.

## 4.2 Candidate Alteration Selection

In the second part, we find candidate alterations that satisfy the performance constraint in Eq. (6). The constraint cannot be accurately computed as it takes an expectation over discrete $G$ and continuous $\boldsymbol{\theta}$. To reduce the computational burden, we estimate the constraint with posterior samples. Note that the value of variables $\mathbf{V}$ is uniquely determined by $G$, $\boldsymbol{\theta}$, and the noise $\boldsymbol{\varepsilon}$ in Eq. (1). We can evaluate all alterations by generating $\mathbf{Y}_t$ from a common set of SRMs and noise samples. Suppose we have an i.i.d. sample $S = \{\langle G_i, \boldsymbol{\theta}_i, \boldsymbol{\varepsilon}_i \rangle\}_{i=1}^n$, the empirical estimate of Eq. (6) is

$$\hat{p}_{\xi_t}(S) \triangleq |\{i \mid \mathbf{y}_{t,\xi_t}^i \in \mathcal{S}\}|/n,$$

where $\mathbf{y}_{t,\xi_t}^i$ is the concerned outcome generated following the SRM specified by $\langle G_i, \boldsymbol{\theta}_i, \boldsymbol{\varepsilon}_i \rangle$ and $Rh(\xi_t)$. A general method for candidate alteration selection has three stages:

1. Sample two sets of i.i.d. parameters, the training set $S_{tr} = \{\langle G_i, \boldsymbol{\theta}_i, \boldsymbol{\varepsilon}_i \rangle\}_{i=1}^n$ and the validation set $S_{val} = \{\langle G_i, \boldsymbol{\theta}_i, \boldsymbol{\varepsilon}_i \rangle\}_{i=n+1}^{2n}$, from $\mathbb{P}(G, \boldsymbol{\theta} \mid D_t)$ and $\mathbb{P}(\boldsymbol{\varepsilon} \mid \boldsymbol{\theta})$.
2. Build a candidate alteration set $\mathcal{C}$ by conducting rehearsals on $S_{tr}$ and finding alterations $\xi_t$ that achieve required performance $\tau$, i.e., $\forall \xi_t \in \mathcal{C}, \hat{p}_{\xi_t}(S_{tr}) \geq \tau$.
3. Validate alterations in $\mathcal{C}$ by rehearsals on $S_{val}$ and remove those with $\hat{p}_{\xi_t}(S_{val}) < \tau$ from $\mathcal{C}$.

The validation procedure is to alleviate overfitting as in standard machine learning pipelines. If stage 2 or 3 outputs an empty set, the system should refuse to make any recommendations. The agent can lower $\tau$ and restart if a smaller $\tau$ is still adequate for the task at hand.

For a given alteration, validation is relatively easy to perform. The only issue to address is to find a set of alterations that pass the test on $S_{tr}$. An immediate approach would be enumerating possible alterations and making rehearsals on training samples to check the constraint. But the enumeration can be computationally prohibitive as the alteration space can be combinatorial and continuous. A compromise between the optimality of recommended alterations and the computational feasibility is to work on some finite subset of alterations using some discretization tricks or heuristics. Fortunately, due to the linear structure of the SRMs considered here, Prop. 5 shows that the outcome variables are linear in the alteration values, implying the existence of more efficient and effective solutions.

**Proposition 5.** *Let $\xi = \{(Z_{a_i}, z_{a_i})\}_{i=1}^k$ be a valid alteration where $Z_{a_i} \in \mathbf{Z}_t$ and $z_{a_i} \in \Delta(Z_{a_i})$. Given $\langle G, \boldsymbol{\theta}, \boldsymbol{\varepsilon} \rangle$ and $\mathbf{x}_t$, the outcome variables are linear in the alteration values. We have*

$$\mathbf{Y}_t = \mathbf{A}\mathbf{x}_t + \mathbf{B}\mathbf{z}^\xi + \mathbf{C}\boldsymbol{\varepsilon}, \tag{8}$$

*where $\mathbf{z}^\xi = (z_{a_1}, \ldots, z_{a_k})^T$. $\mathbf{A}$, $\mathbf{B}$, and $\mathbf{C}$ are constant matrices of appropriate shapes and are uniquely determined by $G$, $\boldsymbol{\theta}$ and altered variables $\mathbf{Z}^\xi = \{Z_{a_i}\}_{i=1}^k$.*

**Size-1 Alteration.** We consider cases where an agent can alter only one variable in each round, *i.e.*, $k = 1$. Suppose the altered variable in round $t$ is $Z_i$. Provided with a fixed SRM, along with fixed noise, Eq. (8) reduces to a line in $\mathbb{R}^{|\mathbf{Y}|}$ with direction $\mathbf{B} \in \mathbb{R}^{|\mathbf{Y}|}$. Due to the convexity of the desired region $\mathcal{S}$, if altering $Z_i$ can make $\mathbf{Y} \in \mathcal{S}$, then the set of feasible alteration values can be represented by an interval $[u, v]$ (the endpoints can be infinity). Finding candidate alterations on the training data is then equivalent to finding elements that intersect at least $n \cdot \tau$ intervals obtained from samples in $S_{tr}$, which is easy to compute. Alg. 1 includes the basic training and validation stages of finding candidate alterations. The overall time complexity is $O(|\mathbf{Z}_t| \cdot n \log n)$. A more detailed algorithm and running time analysis are given in Appendix B.

---

**Algorithm 1** Finding candidate alterations of size 1

**Input:** $S_{tr} = \{\langle G_i, \boldsymbol{\theta}_i, \boldsymbol{\varepsilon}_i \rangle\}_{i=1}^n$, $S_{val} = \{\langle G_j, \boldsymbol{\theta}_j, \boldsymbol{\varepsilon}_j \rangle\}_j$
1: $\mathcal{C} \leftarrow \emptyset$          ▷ Store candidate alterations
2: **for** $Z_i \in \mathbf{Z}$ **do**
3:      $\texttt{int1} \leftarrow \text{FIND}(S_{tr}, Z_i, \tau)$      ▷ Training
4:      $\texttt{int2} \leftarrow \text{FIND}(S_{val}, Z_i, \tau)$
5:      $\mathcal{C} \leftarrow \mathcal{C} \cup \{\langle Z_i, \texttt{int1} \cap \texttt{int2} \rangle\}$    ▷ Validation
6: **function** FIND$(S, Z, \tau)$    ▷ Find alteration values for $Z$
7:      $\texttt{int} \leftarrow \emptyset$          ▷ Store valid intervals
8:      **for** $\langle G, \boldsymbol{\theta}, \boldsymbol{\varepsilon} \rangle \in S$ **do**
9:          $[u, v] \leftarrow$ solution of $\mathbf{M}(\mathbf{Ax} + \mathbf{Bz}^\xi + \mathbf{C}\boldsymbol{\varepsilon}) \leq \mathbf{d}$
10:            ▷ $\mathbf{A}$, $\mathbf{B}$, $\mathbf{C}$ are determined by $G$, $\boldsymbol{\theta}$, $Z$
11:          $\texttt{int} \leftarrow \texttt{int} \cup \{[u, v] \cap \Delta(Z)\}$
12:      **return** $\{z \mid n\tau \leq \text{\#intervals in } \texttt{int} \text{ containing } z\}$

**Output:** The found alterations $\mathcal{C}$

---

**Size-$k$ Alteration.** By relating to the NP-hard maximum feasible subsystem problem [21], we prove Thm. 6, showing that finding a single $\xi$ that satisfies the constraint, which is much simpler

than finding all valid alterations, is difficult if P = NP. This result suggests that we should focus on approximate solutions rather than exact ones. We also give a mixed integer linear programming approach in Appendix B.

**Theorem 6.** *Unless* P = NP, *finding a $k$-dimensional $\mathbf{z}^\xi \in \Delta(\mathbf{Z}^\xi)$ that satisfies $\sum_{i=1}^n \mathbb{I}(\mathbf{M}(\mathbf{A}_i\mathbf{x} + \mathbf{B}_i\mathbf{z}^\xi + \mathbf{C}_i\boldsymbol{\varepsilon}_i) \leq \mathbf{d}) \geq n \cdot \tau$ (if there is a valid solution) is not solvable with any algorithm of running time polynomial in $k$ and $n$.*

### 4.3 Mutual Information Maximization

The remaining task is to optimize the mutual information (5) over the found candidate alteration set $\mathcal{C}$. We factorize the objective function with two information entropy terms over $\mathbf{Z}_t$ and $\mathbf{Y}_t$:

$$\mathrm{I}\left(\mathbf{G}, \boldsymbol{\Theta}; \mathbf{Z}_t, \mathbf{Y}_t \mid D_t, \mathbf{x}_t, \xi_t\right) = \mathrm{H}(\mathbf{Z}_t, \mathbf{Y}_t \mid D_t, \mathbf{x}_t, \xi_t) - \mathrm{H}(\mathbf{Z}_t, \mathbf{Y}_t \mid \mathbf{G}, \boldsymbol{\Theta}, D_t, \mathbf{x}_t, \xi_t). \quad (9)$$

Eq. (9) can be estimated with samples from $\mathbb{P}(G, \boldsymbol{\theta} \mid D_t)$ and $\mathbb{P}(\mathbf{z}_t, \mathbf{y}_t \mid G, \boldsymbol{\theta}, \mathbf{x}_t, \xi_t)$. Due to the sampling involved in the estimation procedure, Eq. (9) is expensive to evaluate. The continuous input space makes it difficult to find an optimal alteration value for a fixed alteration target $\mathbf{Z}^\xi$. Thus, we find $\mathbf{z}^{\xi*}$ with Bayesian optimization [22], which seeks to optimize a black-box function $g(\mathbf{z})$ with a small number of evaluations by iteratively querying function values at some data points. We run the optimization procedure for every alteration target in $\mathcal{C}$ and find the corresponding best alteration values. Finally, the alteration that gives maximum mutual information is selected as the recommended output. Details about the estimation procedure and Bayesian optimization are given in Appendix C.

Given a decision, it is important to know how likely the decision can successfully avoid the undesired future. Though the candidate selection procedure already rules out potentially terrible alterations using $S_{tr}$ and $S_{val}$, a more refined measure is preferable. Based on the scenario approach in robust control literature [23, 24], we derive a PAC-style bound on the posterior success probability in Thm. 7. The bound is easy to compute as it only requires posterior samples. The practical implication of the bound is that it quantifies the uncertainty, or risk, associated with the alteration, thus can help make better decisions. Notably, the bound holds for any SRM with arbitrary additive noise, which could be helpful for future extensions with non-Gaussian noise and nonlinear SRMs.

**Theorem 7.** *Given an observed evidence $\mathbf{x}_t$, an alteration $\xi$, and a desired set $\mathcal{S} = \{\mathbf{y} \in \mathbb{R}^{|\mathbf{Y}|} \mid \mathbf{M}\mathbf{y} \leq \mathbf{d}\}$, let $S_{eval} = \{\langle G_i, \boldsymbol{\theta}_i, \boldsymbol{\varepsilon}_i \rangle\}_{i=1}^n$ be $n$ i.i.d. samples from $\mathbb{P}(G, \boldsymbol{\theta}, \boldsymbol{\varepsilon} \mid D_t)$. Let $\{\mathbf{y}^i\}_{i=1}^n$ be generated from $Rh(\xi)$ and $S_{eval}$ following the structural equations in Eq. (1). Define $n_o \triangleq |\{i \mid \mathbf{y}^i \notin \mathcal{S}\}|$. Let $\hat{p} \triangleq 1 - n_o/n$, $\underline{p} \triangleq 1 - F^{-1}(1 - \delta/(2n); n_o + 1, n - n_o)$ if $n_o < n$ otherwise $\underline{p} \triangleq 0$, and $\bar{p} \triangleq 1 - F^{-1}(\delta/(2n); n_o, n - n_o + 1)$ if $n_o > 0$ otherwise $\bar{p} \triangleq 1$, where $F^{-1}(\,\cdot\,; \alpha, \beta)$ is the inverse cumulative distribution function of the beta distribution with parameters $\alpha$ and $\beta$. For any $\delta \in (0, 1)$, with probability at least $1 - \delta$, we have*

$$\max\left\{\underline{p}, \hat{p} - \sqrt{\ln(2/\delta)/2n}\right\} \leq \mathbb{P}\left(\mathbf{Y}_t \in \mathcal{S} \mid D_t, \mathbf{x}_t, Rh(\xi)\right) \leq \min\left\{\bar{p}, \hat{p} + \sqrt{\ln(2/\delta)/2n}\right\}.$$

### 4.4 Discussion

The proposed rehearsal learning framework is general and flexible, with any of the three steps adjustable without interfering with the others. For scenarios more complicated than linear ones, one can replace methods in each step accordingly: The Bayesian update of SRMs can use any advanced Bayesian (structure) learning methods [25, 26, 27, 28] as long as they allow sampling from the posterior. The mutual information maximization step can leverage any black-box optimization methods [22, 29]. For the most important and difficult candidate alteration finding step, which is difficult even for the linear case when allowing joint alterations, efficient heuristics or linearization tricks can be used. Given the great uncertainty of real-world problems, exactly solving the AUF problem is extremely difficult if not impossible, while with the rehearsal learning framework, we provide a practical and systematic approximate solution that tackles the problem with certain guarantees, which exhibits that developing more efficient approximate solutions is a promising future direction.

For the SRM, there are some practical considerations. After a rehearsal operation, the change of the parental relations results in new sets of parameters. In the absence of further constraints, as is the case in this work, these new parameters may bear no direct resemblance to the previous ones. However, it is conceivable to impose specific assumptions on these parameters to achieve more

efficient modeling. For instance, one may assume the new parameters are connected with the original ones through operations like marginalization. In practical implementations, it is also reasonable to limit the size of a clique and maintain the number of parameter sets to an acceptable level. Moreover, it is unlikely that we would need to or be able to alter every possible subset of a clique, so one can reduce unnecessary computational burden by ignoring the parameters that would not be used.

## 5 Related Work

There have been some efforts to combine structural models with decision-making, with most of them focusing on causal structures [30]. Some work explores the causal bandit problem, where the causal structure underlying rewards and actions is considered [31, 32, 33, 34, 35, 36]. In a different vein, Aglietti et al. [37] investigated a Bayesian optimization problem with a known causal DAG, and Aglietti et al. [38] extended this work to a dynamic setting. However, these studies typically assume that the true causal graph is known, a condition that is challenging to meet in real-world applications, although causal discovery methods could help to some extent [39, 40, 41, 42, 43, 44]. To remove the assumption, de Kroon et al. [45] leveraged separating sets, while Lu et al. [46] focused on the Markov equivalence class of the true causal graphs. However, even when causal relations are known, applying causal bandits or causal Bayesian optimization to AUF is not suitable because they often seek a single universally optimal action. In contrast, optimal decisions in AUF can vary depending on the context $\mathbf{X}$. Some research approaches decision problems from the perspective of causal structures and causal effect estimation. For example, Wang et al. [47] developed a method for estimating a set of possible causal effects to aid decision-making. Additionally, there has been active research in estimating causal structures or effects in interactive environments [48, 49, 50, 51, 52].

The aforementioned efforts yielded effective methods for certain decision-related problems, but all of them rely on causal modeling, which could be too luxurious and restrictive for decision problems [7]. On the other hand, correlation, which is the basis of most ML models, is insufficient for decision-making. Therefore, we turn to the *influence relation*, which lies between correlation and causation and forms a basis of decisions [7]. Building on the influence relation, we propose SRM, which is capable of modeling interrelated but not necessarily causally linked variables and dynamically evolving decision systems. Moreover, the proposed rehearsal learning framework demonstrates the feasibility of building decision-making upon influence and rehearsal.

## 6 Experiments

We evaluate the proposed approach on two datasets. We are mainly interested in if the proposed approach can successfully avoid the undesired future with high probability, the exploration-exploitation trade-off, and the informativeness of the PAC-style guarantees. For each dataset, we alter one variable in each round and repeat experiments with 100 rounds 20 times. The graph prior is initialized with samples from the learnable graph equivalence class instead of learning from data as well-established rehearsal graph learning methods are still in the process of development. The success probability is estimated with 1,000 samples from the true SRM. We compute the PAC bound with 1,000 samples from the posterior with $\delta = 0.05$. The observational dataset size is set to 10. We also compare with several reinforcement learning methods DDPG [53], PPO [54], SAC [55], and CATS [56].

**Ride-Hailing Data.** We abstract an SRM from a ride-hailing scenario, where a ride-hailing app needs to make decisions to promote user rating (RAT). We consider relevant variables including the weather condition, the number of users, traffic congestion (TRA), etc. The app can alter two variables, the discount level (DIS) and the recommendation level (REC) of a specific route. There exist interrelated variables in the scenario: If REC of a specific route is high, then TRA on that route will be high since more users will choose that route; and if there is a high TRA on that route, the REC should be low. The size of $\mathbf{X}_t$, $\mathbf{Z}_t$, and $\mathbf{Y}_t$ are 2, 4, and 1 respectively. The true structural equations are set according to domain knowledge. The feasible alteration values are $[-2, 2]$ for DIS and REC. The range of RAT is $[0, 1]$, so we set the desired region to $\mathcal{S} = [0.8, 1]$ to avoid RAT below $0.8$.

**Bermuda Data.** We take an example from ecology, where environment variables in Bermuda are recorded [57] and the variable generation order is available [58]. The size of $\mathbf{X}_t$, $\mathbf{Z}_t$, $\mathbf{Y}_t$ are 3, 7, and 1 respectively. The true structural equations are obtained by performing linear regression on

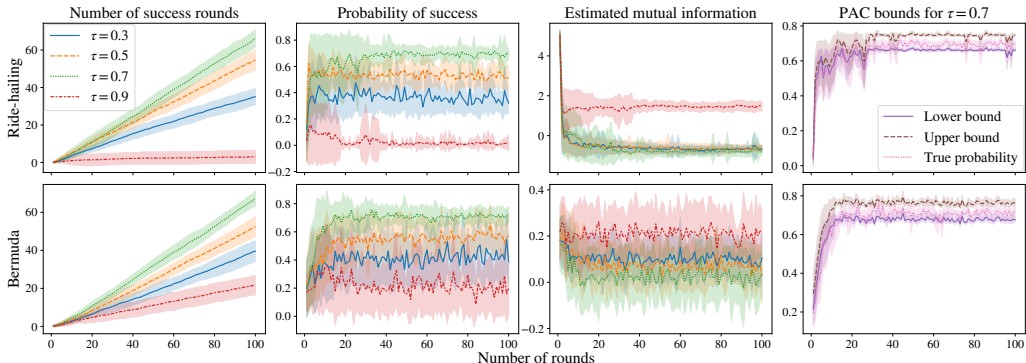

Figure 4: Results on the Ride-hailing and the Bermuda data. The bands depict standard deviations.

normalized data. We assume that 5 variables are actionable [37], with a feasible alteration region $[-1, 1]$. $\mathbf{Y}_t$ represents the net coral ecosystem calcification (NEC). We want to maintain a high NEC. So we set the desired region to $\mathcal{S} = [0.5, 2]$ which is above the 75th percentile of the dataset.

Table 1 shows the average probabilities of successfully avoiding the undesired future of several RL methods and the proposed rehearsal learning method with $\tau = 0.7$. Our method achieves the goal of AUF with a probability around 0.7, while the others fail to do so with merely 100 interac-

| Dataset | DDPG | PPO | SAC | CATS | $\tau = 0.7$ |
|---|---|---|---|---|---|
| Ride-hailing | 0.173 | 0.154 | 0.177 | 0.104 | 0.714 |
| Bermuda | 0.230 | 0.190 | 0.205 | 0.215 | 0.679 |

Table 1: Avg. success probability.

tions with the environment, which further underscores the importance of considering structural information in the interaction-limited AUF problem. On the other hand, if we increase $T$, RL methods can achieve satisfying performance, e.g., DDPG achieves 0.688 average success probability when $T = 10,000$ on the Bermuda data. For the ride-hailing data, 95.8% alterations recommended by the proposed method are to alter REC, which affects RAT through the interactions between REC and TRA and the direct link between TRA and REC: REC $\leftrightarrow$ TRA $\rightarrow$ RAT. Note that REC cannot be identified as a valid cause so causal modeling is not suitable for this problem.

Fig. 4 shows the full results. Given $\tau$, the found alteration can satisfy the probability requirement in most cases. Exceptions are settings where $\tau = 0.9$. Alterations with such a high success probability rarely exist, so the method can fail. In our implementation, when the recommendation is not attainable, the constraint is dropped and the method will only optimize the mutual information. Therefore the corresponding number of success rounds, as well as the success probability, is low, but the mutual information is high. The third column shows a trade-off between avoiding the undesired future and learning the SRM: smaller $\tau$ is likely to give higher mutual information. An interesting phenomenon is that the true success probability often fluctuates around $\tau$ instead of achieving higher values. An explanation is that if an alteration has a high success probability, then the effect of the alteration is less uncertain, so less new information can be revealed and the method will instead choose others with lower success probabilities. Finally, the last column shows that the true probabilities are bounded by the PAC-style bound with a relatively small gap, so it can be informative for guiding real decisions.

## 7 Conclusion

Realizing that *correlation* is inadequate and that *causation* is not always suitable, we advocate relying on the *influence* relation for decision-making. In this paper, we propose the first rehearsal learning framework that tackles the AUF (Avoiding Undesired Future) problem with influence modeling. The key to the proposed framework is SRM, which considers the interrelations of variables and the dynamic time-dependent nature of decision environments. By unifying rehearsals on SRMs and Bayesian inference, the framework recommends decisions that can persuasively avoid the undesired future. We also provide a PAC-style bound to quantify the associated risk of recommended decisions and further show the hardness of a basic linear instance of the framework, revealing that future work should focus on developing approximate solutions rather than exact ones.

## Acknowledgments

This research was supported by National Key R&D Program of China (2022ZD0114800) and NSFC (61921006). The authors thank the reviewers for their valuable comments.

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

# A  Rehearsal Graphs: Characterization and Learning

In this section, we present full motivations, basic properties, and preliminary learning strategies for rehearsal graphs.

The ability to describe and identify causal relations is at the heart of both human and artificial intelligence. The work of Pearl [12] on causality lays the foundation for accurately describing causal relations and making a sound inference. Despite its elegance and expressiveness in abstracting causality from human cognition, one may encounter difficulties when applying Pearl's causal framework to real-world observations. We identify three kinds of misspecifications that could make causal modeling fail: a) model misspecification b) data-collection misspecification c) adversary misspecification.

Model misspecification means that the variables used in the causal modeling process are inaccurate so there are indeterminant causal relations among the variables. For example, demand ($D$) and price ($P$) are two variables that are causally related. But there is not a determinant causal direction between $D$ and $P$ since intervening on $D$ causes $P$ to change but meanwhile intervening on $P$ causes $D$ to change. A workaround is modeling with finer granularity by adding a time index to $D$ and $P$. Then we have $D_t$ causes $P_{t+1}$ and $P_t$ causes $D_{t+1}$.

Data-collection misspecification means that the data collected are not perfectly aligned with the specified variable. For example, we may have variables $D_1, D_2, P_1, P_2$ in the causal model that denote two measures $D$ and $P$ at different time. But the data collected may not rigorously follow the order since there can be time delays and mismatches of the modeling time intervals and real ones. This misspecification results in a misalignment between the model and data, hence no sound causal conclusions can be drawn from the data.

Adversary misspecification refers to ignoring the existence of adversaries in causal modeling. For example, let $P$ and $Q$ denote the prices of a product in two stores. If $P$ is decreased then $Q$ will also be decreased, otherwise, the second store will lose some consumers. From a causal view, it seems that $P$ causes $Q$. But obviously, $Q$ also causes $P$ by symmetry. So we have a bi-directional causal relation, which is cannot be described in causal modeling. The reason for this phenomenon is that each store is an adversary that reacts to interventions from other agents.

The preceding three misspecifications are not necessarily disjoint in real problems and are hard to identify in advance. The consequence is that one cannot identify perfect causal knowledge even if with an infinite amount of data since the modeling and the data do not exhibit consistent determinant causal relations. In order to model both the non-determinant and determinant causal relations in data, we introduce a new probabilistic graphical model: rehearsal graphs.

## A.1  Rehearsal Graphs as a Probabilistic Graphical Model

**Definition 8** (Mixed graph). Let $G = (\mathbf{V}, \mathbf{E})$ be a graph, where $\mathbf{V}$ denotes the vertices and $\mathbf{E}$ the edges. $G$ is a *mixed graph* if for any distinct vertices $u, v \in \mathbf{V}$, there is at most one edge connecting them, and the edge can be either directional or bi-directional, *i.e.*, $u \to v$, $u \leftarrow v$, or $u \leftrightarrow v$.

**Definition 9** (Bi-directional clique). A *bi-directional clique* $C = (\mathbf{V}^c, \mathbf{E}^c)$ of a mixed graph $G = (\mathbf{V}, \mathbf{E})$ is a complete induced subgraph such that any edge $e \in \mathbf{E}^c$ is bi-directional. A single vertex also constituents a valid bi-directional clique. $C$ is *maximal* if adding any other vertex does not induce a bi-directional clique.

**Definition 10** (Rehearsal graph). Let $G = (\mathbf{V}, \mathbf{E})$ be a mixed graph. Let $\{C_i\}_{i=1}^l$ denote all maximal bi-directional cliques of $G$, where $C_i = (\mathbf{V}_i^c, \mathbf{E}_i^c)$. $G$ is a *rehearsal graph* if and only if:

1. $\mathbf{V}_i^c \cap \mathbf{V}_j^c = \emptyset$ for any $i \neq j$.
2. $\forall i \in [l]$, $u \in \mathbf{V} \setminus \mathbf{V}_i^c$, if there is any edge pointing from $u$ to $\mathbf{V}_i^c$, then $\forall v \in \mathbf{V}_i^c$, $u \to v$.
3. The directional edges permit a topological ordering for $\{C_i\}_{i=1}^l$.

**Definition 11** (Factorization). Let $G = (\mathbf{V}, \mathbf{E})$ be a rehearsal graph and $C_i = (\mathbf{V}_i^c, \mathbf{E}_i^c), i \in [k]$ be maximal bi-directional cliques of $G$. We say that a joint probability $P$ over $\mathbf{V}$ factorizes according to $G$ if $P$ can be expressed as a product

$$P(\mathbf{V}) = \prod_{i=1}^l P(\mathbf{V}_i^c \mid \mathbf{PA}_i), \tag{10}$$

where $\mathbf{PA}_i = \{u \mid \exists v \in \mathbf{V}_i^c, u \to v \text{ in } G\}$.

## A.2  $c$-Separation

As a probabilistic graphical model, like Bayesian networks or Markov networks, rehearsal graphs also have a graphical characterization of independence relations. Mimicking the $d$-separation for Bayesian networks, we present $c$-separation for rehearsal graphs. Here the "$c$" means "cliques".

**Definition 12** ($c$-separation). A path $p$ in a rehearsal graph $G$ is said to be $c$-separated (or blocked) by a set of vertices $Z$ if and only if

1. $p$ contains a chain $i \to m_1 \leftrightarrow m_2 \leftrightarrow \cdots \leftrightarrow m_l \to j$ or a fork $i \leftarrow m_1 \leftrightarrow m_2 \leftrightarrow \cdots \leftrightarrow m_l \to j$ such that the bi-directionally connected vertices $\{m_i\}_{i=1}^l \subseteq Z$, or

2. $p$ contains an inverted fork (or collider) $i \to m_1 \leftrightarrow m_2 \leftrightarrow \cdots \leftrightarrow m_l \leftarrow j$ such that no vertices in $\{m_i\}_{i=1}^l$ are in $Z$ and such that no descendants of $\{m_i\}_{i=1}^l$ are in $Z$.

A set $Z$ is said to $c$-separate $X$ from $Y$ if and only if $Z$ blocks every path from a vertex in $X$ to a vertex in $Z$. We write $X \perp\!\!\!\perp_G Y \mid Z$ when $Z$ $c$-separates $X$ from $Y$ in $G$.

**Lemma 13.** *Let $G = (\mathbf{V}, \mathbf{E})$ be a rehearsal graph, $x, y \in \mathbf{V}$ be two distinct vertices, and $Z$ a set of vertices. Let $C(v)$ denote the vertices of the maximal bi-directional clique to which $v$ belongs. If $x \perp\!\!\!\perp_G y \mid Z$, then $C(x) \perp\!\!\!\perp_G C(y) \mid Z$.*

*Proof.* We prove this by contradiction. Assume that $C(x) \perp\!\!\!\perp_G C(y) \mid Z$ does not hold given $x \perp\!\!\!\perp_G y \mid Z$, meaning that there exists a path $p$ unblocked by $Z$ between some $u \in C(x)$ and some $v \in C(y)$. Then there is an unblocked path $x \leftrightarrow p \leftrightarrow y$ between $x$ and $y$, which is a contradiction since $Z$ is supposed to block every path between $x$ and $y$. □

**Lemma 14.** *Let $G = (\mathbf{V}, \mathbf{E})$ be a rehearsal graph, $C_a, C_b \subseteq V$ be a set of vertices of two distinct maximal bi-directional cliques, and $Z$ a set of vertices. If $C_a \perp\!\!\!\perp_G C_b \mid Z$, then for all probability functions $P$ that factorizes according to $G$, $C_a \perp\!\!\!\perp_P C_b \mid Z$.*

*Proof.* We construct a DAG $\tilde{G} = (\tilde{V}, \tilde{E})$ from $G$ with an one-to-one mapping $f$: every maximal cliques $C_i$ of $G$ is represented with a vertex $\tilde{v}_i \in \tilde{G}$, $\tilde{v}_i = (v_i^1, \ldots, v_i^{|C_i|})$, and the edge directions between $C_i$s are kept. For any path $p$ $\tilde{v}_{\alpha_1} - \tilde{v}_{\alpha_2} - \cdots - \tilde{v}_{\alpha_l}$ (here we omit the edge directions) in $\tilde{G}$, where $\{\alpha_i\}_i$ is an index set, there is a counterpart $q$ in the original $G$ that reads $v_{\alpha_1}^1 \leftrightarrow v_{\alpha_1}^2 \leftrightarrow \cdots \leftrightarrow v_{\alpha_1}^{|C_{\alpha_1}|} - v_{\alpha_2}^1 \leftrightarrow v_{\alpha_2}^2 \leftrightarrow \cdots \leftrightarrow v_{\alpha_2}^{|C_{\alpha_2}|} - \cdots - v_{\alpha_l}^1 \leftrightarrow v_{\alpha_l}^2 \leftrightarrow \cdots \leftrightarrow v_{\alpha_l}^{|C_{\alpha_l}|}$, where $v_{\alpha_k}^j$ is the $j$-th vertex in $C_{\alpha_k}$. For any $Z \subseteq V$, we map it to a set of vertices in $\tilde{G}$ with $h(Z) = \{\tilde{v} \in \tilde{V} \mid f^{-1}(\tilde{v}) \subseteq Z\}$. From the definition of $c$-separation for rehearsal graphs and $d$-separation for Bayesian networks, we see that if $q$ is $c$-separated by $Z$ in $G$, then $p$ is $d$-separated by $h(Z)$ in $\tilde{G}$. Therefore, if $C_a \perp\!\!\!\perp_G C_b \mid Z$, then every path between $\tilde{v_a}$ and $\tilde{v}_b$ is $d$-separated by $h(Z)$. Due to the Markov property of Bayesian networks, for all $P$ that factorizes according to $\tilde{G}$ (which also factorizes according to $G$ by definition), we have $\tilde{v}_a \perp\!\!\!\perp_P \tilde{v}_b \mid h(Z)$, *i.e.*, $C_a \perp\!\!\!\perp_P C_b \mid f^{-1}(h(Z))$. Since $f^{-1}(h(Z)) \subseteq Z$, we have $C_a \perp\!\!\!\perp_P C_b \mid Z$. □

**Theorem 15.** *Let $G = (\mathbf{V}, \mathbf{E})$ be a rehearsal graph. For all probability functions $P$ that factorizes according to $G$ and any three disjoint subsets of vertices $X, Y, Z \subseteq \mathbf{V}$, if $X \perp\!\!\!\perp_G Y \mid Z$, then $X \perp\!\!\!\perp_P Y \mid Z$.*

*Proof.*

$$X \perp\!\!\!\perp_G Y \mid Z \quad \Rightarrow \quad \forall x \in X, y \in Y, x \perp\!\!\!\perp_G y \mid Z \tag{11}$$

$$\xRightarrow{\text{Lem. 13}} \quad \forall x \in X, y \in Y, C(x) \perp\!\!\!\perp_G C(y) \mid Z \tag{12}$$

$$\xRightarrow{\text{Lem. 14}} \quad \forall x \in X, y \in Y, C(x) \perp\!\!\!\perp_P C(y) \mid Z \tag{13}$$

$$\Rightarrow \quad \forall x \in X, y \in Y, x \perp\!\!\!\perp_P y \mid Z \tag{14}$$

$$\Rightarrow \quad X \perp\!\!\!\perp_P Y \mid Z. \tag{15}$$

□

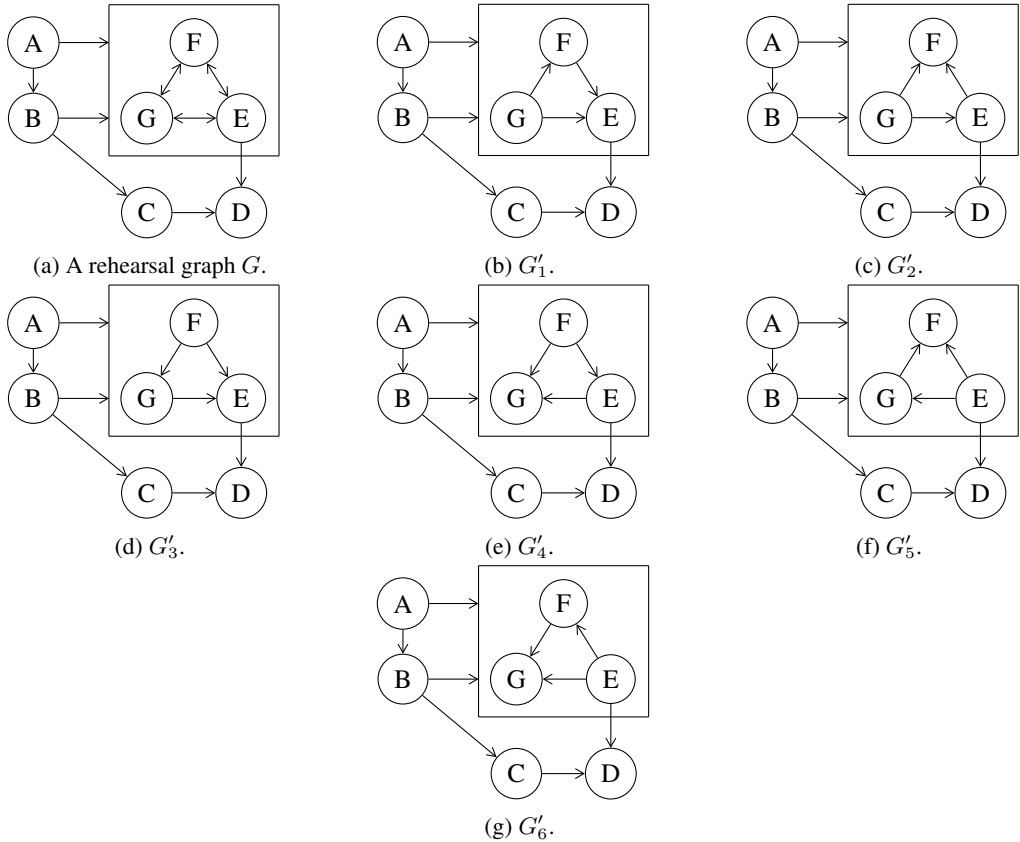

(a) A rehearsal graph $G$.     (b) $G_1'$.     (c) $G_2'$.

(d) $G_3'$.     (e) $G_4'$.     (f) $G_5'$.

(g) $G_6'$.

Figure 5: An example rehearsal graph and its Bayesian network counterparts that encode the same conditional independence relations. We use plate notation to simplify the edges. For example, in (a), vertex A having an edge pointing to the rectangle containing three vertices, implies that A→E, A→F, and A→G.

**Theorem 16.** *Let $G = (\mathbf{V}, \mathbf{E})$ be a rehearsal graph. Let $G' = (\mathbf{V}, \mathbf{E}')$ be any DAG that have identical skeleton with $G$ (see Fig. 5 for an example). If all directional edges in $G$ appear in $G'$, then for any disjoint set of vertices $X, Y$ and $Z$, we have $X \perp\!\!\!\perp_G Y \mid Z$ if and only if $X \perp\!\!\!\perp_{G'} Y \mid Z$, where $\perp\!\!\!\perp_G$ means c-separation for rehearsal graphs and $\perp\!\!\!\perp_{G'}$ d-separation for DAGs.*

*Proof.* We first prove $X \perp\!\!\!\perp_G Y \mid Z \Rightarrow X \perp\!\!\!\perp_{G'} Y \mid Z$.

Every path $p$ in $G$ has a unique counterpart $p'$ in $G'$ that shares the same vertex sequence. $p$ and $p'$ only differ in some edges: a bi-directional edge $u \leftrightarrow v$ in $G$ is a directed one $u \to v$ or $u \leftarrow v$ in $G'$. Obviously, the mapping between $p$ and $p'$ is one-to-one. We only need to show that for any path $p'$ in $G'$, if its counterpart $p$ in $G$ is c-separated by a set $Z$, then $p'$ is d-separated in $G$.

We prove it by contradiction. Assume that $p'$ is not d-separated by $Z$, then according to the definition of d-separation, we have that (a) for every chain $i \to m \to j$ and every fork $i \leftrightarrow m \leftarrow j$ in $p'$, $m \notin Z$, and that (b) for every inverted fork $i \to m \leftarrow j$ in $p'$, $(\{m\} \cup des(m)) \cap Z \neq \emptyset$. For every chain $i \to m_1 \leftrightarrow m_2 \leftrightarrow \cdots \leftrightarrow m_l \to j$ and every fork $i \leftarrow m_1 \leftrightarrow m_2 \leftrightarrow \cdots \leftrightarrow m_l \to j$ in $p$, we have $\{m_i\}_{i=1}^l \not\subseteq Z$ since otherwise either $* \leftarrow m_l \to j$ in $p'$ and $m_l \in Z$, or $* \to m_l \to j$ in $p'$ and $m_l \in Z$, which violate requirement (a). For every inverted fork $i \to m_1 \leftrightarrow m_2 \leftrightarrow \cdots \leftrightarrow m_l \leftarrow j$ in $p$, $(\{m_i\}_{i=1}^l \cup des(\{m_i\}_{i=1}^l)) \cap Z \neq \emptyset$ since otherwise there exists an inverted fork $* \to m_t \leftarrow *, t \in [l]$ such that $(\{m_t\} \cup des(m_t)) \cap Z = \emptyset$, which violates requirement (b). Combining the above results, we have that $p$ is not c-separated by $Z$, which is a contradiction. Therefore, $p'$ is d-separated by $Z$, which concludes that $X \perp\!\!\!\perp_G Y \mid Z \Rightarrow X \perp\!\!\!\perp_{G'} Y \mid Z$.

We next prove $X \perp\!\!\!\perp_{G'} Y \mid Z \Rightarrow X \perp\!\!\!\perp_G Y \mid Z$.

Again we prove by contradiction. Assume that there exists a path $p$ from $x \in X$ to $y \in Y$ in $G$ that is not $c$-separated by $Z$. Let $p$ be $x - m_1^1 \leftrightarrow m_1^2 \leftrightarrow \cdots \leftrightarrow m_1^{l_1} - m_2^1 \leftrightarrow m_2^2 \leftrightarrow \cdots \leftrightarrow m_2^{l_2} - \cdots - m_t^1 \leftrightarrow m_t^2 \leftrightarrow \cdots \leftrightarrow m_t^{l_t} - y$ (we omit the direction of directional edges). We construct a path $q$ from $x$ to $y$ by picking one vertex from each bi-directionally connected subpath $m_i^1 \leftrightarrow m_i^2 \leftrightarrow \cdots \leftrightarrow m_i^{l_i}$ in order such that $q$ contains only directional edges, so q is also a path in $G'$. For each bi-directionally connected subpath $m_i^1 \leftrightarrow m_i^2 \leftrightarrow \cdots \leftrightarrow m_i^{l_i}$ in $p$, if it is a part of a chain $* \to m_i^1 \leftrightarrow m_i^2 \leftrightarrow \cdots \leftrightarrow m_i^{l_i} \to *$ or a fork $* \leftarrow m_i^1 \leftrightarrow m_i^2 \leftrightarrow \cdots \leftrightarrow m_i^{l_i} \to *$, then by the definition of $c$-separation, $\{m_i^j\}_{j=1}^{l_i} \setminus Z \neq \emptyset$, so we randomly select a vertex $v_i \in \{m_i^j\}_{j=1}^{l_i} \setminus Z$ to join $q$. If the bi-directionally connected subpath is a part of an inverted fork $* \to m_i^1 \leftrightarrow m_i^2 \leftrightarrow \cdots \leftrightarrow m_i^{l_i} \leftarrow *$, then by the definition of $c$-separation, either some $m_i^j \in Z$ or some descendant of $\{m_i^j\}_{j=1}^{l_i}$ is in $Z$. In the former case, we select $v_i = m_i^j$ to join $q$, and in the latter, we randomly select any $v_i \in \{m_i^j\}_{j=1}^{l_i}$. By construction, $q = x - v_1 - \cdots - v_t - y$ is a path from $X$ to $Y$ in $G'$ and is not $d$-separated, which is a contradiction. Therefore, any path $p$ from $X$ to $Y$ in $G$ is $c$-separated by $Z$, which concludes $X \perp\!\!\!\perp_{G'} Y \mid Z \implies X \perp\!\!\!\perp_G Y \mid Z$. $\qquad \square$

**Theorem 17.** *Let $G = (\mathbf{V}, \mathbf{E})$ be a rehearsal graph. For any three disjoint subsets of vertices $X, Y, Z \subseteq V$, if $X \perp\!\!\!\perp_P Y \mid Z$ for all probability functions $P$ that factorizes according to $G$, then $X \perp\!\!\!\perp_G Y \mid Z$.*

*Proof.* Let $G' = (\mathbf{V}, \mathbf{E}')$ be any DAG that has identical skeleton with $G$ and has all directional edges in $G$. Let $V_i = \{v_j\}_{j=1}^l$ denote the set of vertices in a maximal bi-directional clique in $G$ and $PA_i$ the parent of $V_i$. Then $V_i$ is also a clique in $G'$. Without loss of generality, assume the topological ordering among $V_i$ is $v_1 \prec v_2 \prec \cdots \prec v_l$. By the chain rule, we have $P(V_i \mid PA_i) = \prod_{k=1}^l P(v_k \mid v_{k-1}, v_{k-2}, \dots)$, where the LHS is the conditional probability defined in the factorization of $G$ and the RHS is the factorization defined by $G'$. So if $P$ is a probability function that factorizes according to $G$, then $p$ also factorizes according to $G'$, and vice versa. Let $\mathcal{P}(\cdot)$ denote the probability family that factorizes according to $\cdot$. We have $\mathcal{P}(G) = \mathcal{P}(G')$.

$$\forall P \in \mathcal{P}(G), X \perp\!\!\!\perp_P Y \mid Z \xRightarrow{\mathcal{P}(G) = \mathcal{P}(G')} \forall P \in \mathcal{P}(G'), X \perp\!\!\!\perp_P Y \mid Z \tag{16}$$

$$\xRightarrow{\text{completeness of } d\text{-separation}} X \perp\!\!\!\perp_{G'} Y \mid Z \tag{17}$$

$$\xRightarrow{\text{Thm. 16}} X \perp\!\!\!\perp_G Y \mid Z. \tag{18}$$

$\square$

### A.3 Structure Learning

If we drop the bi-directional edges in a rehearsal graph, it becomes a valid Bayesian network, which means that Bayesian networks are special cases of rehearsal graphs. So learning rehearsal graphs can only be harder than learning Bayesian networks. In this subsection, we present a preliminary learning method. More efficient learning methods are left for further research.

Thm. 16 shows that a rehearsal graph has several Bayesian network counterparts that have the same skeleton and encode the same conditional independence relations. So by learning a Bayesian network or a Markov equivalence class of Bayesian networks from data, we can determine the skeleton of the rehearsal graph. Then we can get a rehearsal graph or a set of rehearsal graphs that cannot be distinguished from observational data by adding bi-directional edges to the edges whose directions are not determined. Assuming that the observational distribution factorizes according to a rehearsal graph and that all conditional independence relations of the distribution are encoded by the rehearsal graph, Algorithm 2 lists all valid rehearsal graphs. The first three steps of Algorithm 2 are simply the IC algorithm for Bayesian network learning [12]. Then we list all Bayesian networks and map them back to rehearsal graphs by replacing directional edges with bi-directional ones.

---

**Algorithm 2** Learning rehearsal graphs from observational data

---

**Input:** Observational distribution $P(\mathbf{V})$.

1: For each pair of variables $a$ and $b$ in $\mathbf{V}$, search for a set $S_{ab}$ such that $a \perp\!\!\!\perp b \mid S_{ab}$ holds. Construct an undirected graph $G$ such that vertices $a$ and $b$ are connected with an edge if and only if no set $S_{ab}$ can be found.

2: For each pair of nonadjacent variables $a$ and $b$ with a common neighbor $c$, check if $c \in S_{ab}$. If it is, then continue. If it is not, then add arrowheads pointing at $c$, *i.e.*, $a \rightarrow c \leftarrow b$.

3: In the partially directed graph that results, orient as many of the undirected edges as possible subject to two conditions:

   (i) Any alternative orientation would yield a new $v$-structure;

  (ii) Any alternative orientation would yield a directed cycle.

4: List all Markov equivalent directed graphs from the partially directed graph. Denote the set of Markov equivalent graphs with $\mathcal{G}_1$.

5: For each listed graph $g$ in $\mathcal{G}_1$, build a set of mixed graphs $\mathcal{G}_2^g$ by enumerating all possible combinations of cliques and replace the directional edges therein with bi-directional edges.

6: Check if the graphs in each $\mathcal{G}_2^g$ are valid rehearsal graphs. If they are not, remove them from corresponding $\mathcal{G}_2^g$.

**Output:** Markov equivalent rehearsal graphs $\mathcal{G}_1 \cup (\cup_{g \in \mathcal{G}_1} \mathcal{G}_2^g)$.

---

# B   Candidate Alteration Selection

## B.1   Size-$1$ Alteration

Algorithm 1 in the main text shows the main steps of finding candidate alterations, where details on finding alteration values that satisfy the performance requirement are omitted. Here we give more details in Algorithm 3. The main idea is to sort the found intervals in increasing order, then keep those that appear more than $n \cdot \tau$ times. Sorting the array in Line 17 on average costs $O(n \log n)$ time if using quicksort. Other operations cost $O(n)$ time. Thus, the overall average running time of Algorithm 3 is $O(|\mathbf{Z}_t| \cdot n \log n)$.

## B.2   Size-$k$ Alteration

We propose to build a finite candidate set by solving a mixed-integer linear programming problem:

$$\max_{\boldsymbol{e} \in \{0,1\}^{n'}, \mathbf{z}^\xi} \quad \sum_i \boldsymbol{e}_i$$

$$\text{s.t.} \ \ \mathbf{M}(\mathbf{A}_i\mathbf{x} + \mathbf{B}_i\mathbf{z}^\xi + \mathbf{C}_i\boldsymbol{\varepsilon}_i) - \mathbf{d} \le (1 - \boldsymbol{e}_i)\boldsymbol{\alpha}, \ i \in [n'],$$

where $\boldsymbol{e}_i$ is a binary decision variable that equals 1 if the $i$-th inequality is satisfied and 0 otherwise, and $\boldsymbol{\alpha}$ is a constant vector that upper bounds the left-hand side. If the found solution satisfies $\sum_i \boldsymbol{e}_i \ge n \cdot \tau$, corresponding $\mathbf{z}^\xi$ will be joined in the candidate set. The above programming is run for every combination of at most $k$ alteration targets multiple times to obtain multiple alterations. The hard constraints satisfied by previous programs for a common $\mathbf{Z}^\xi$ are randomly dropped in following iterations to encourage diverse solutions, thus the $n' \le n$ constraints instead of $n$.

# C   Mutual Information Maximization

Here we detail the estimation of the mutual information objective in Eq. (5). We have

$$\mathrm{I}\left(\mathbf{G}, \boldsymbol{\Theta}; \mathbf{Z}_t, \mathbf{Y}_t \mid D_t, \mathbf{x}_t, \xi_t\right) = \mathrm{H}(\mathbf{Z}_t, \mathbf{Y}_t \mid D_t, \mathbf{x}_t, \xi_t) - \mathrm{H}(\mathbf{Z}_t, \mathbf{Y}_t \mid \mathbf{G}, \boldsymbol{\Theta}, D_t, \mathbf{x}_t, \xi_t).$$

In order to estimate the mutual information, we only need to estimate $\mathrm{H}(\mathbf{Z}_t, \mathbf{Y}_t \mid D_t, \mathbf{x}_t, \xi_t)$ and $\mathrm{H}(\mathbf{Z}_t, \mathbf{Y}_t \mid \mathbf{G}, \boldsymbol{\Theta}, D_t, \mathbf{x}_t, \xi_t)$. We give a detailed estimation procedure in Algorithm 4. The entropy estimation method ENTROPYESTIMATE that estimates information entropy with samples is used as a black-box method since various methods are available, *e.g.*, Sricharan et al. [59].

Let $g \triangleq \mathrm{I}\left(\mathbf{G}, \boldsymbol{\Theta}; \mathbf{Z}_t, \mathbf{Y}_t \mid D_t, \mathbf{x}_t, \xi_t\right)$ denote the target optimization function to be used in Bayesian optimization [22]. A Gaussian process is used to attain the posterior of $g$. Denote the posterior

**Algorithm 3** Find candidate single alterations

---

**Input:** $S_{tr} = \{\langle G_i, \boldsymbol{\theta}_i, \boldsymbol{\varepsilon}_i \rangle\}_{i=1}^{n}, S_{val} = \{\langle G_i, \boldsymbol{\theta}_i, \boldsymbol{\varepsilon}_j \rangle\}_{i=n+1}^{2n}, \tau$

1:   $\mathcal{C} \leftarrow \emptyset$        $\triangleright$ Store candidate alterations
2:   **for** $Z_i \in \mathbf{Z}$ **do**
3:      $\texttt{int1} \leftarrow \text{FIND}(S_{tr}, Z_i, \tau)$        $\triangleright$ Training
4:      $\texttt{int2} \leftarrow \text{FIND}(S_{val}, Z_i, \tau)$
5:      $\mathcal{C} \leftarrow \mathcal{C} \cup \{\langle Z_i, \texttt{int1} \cap \texttt{int2}\rangle\}$        $\triangleright$ Validation
6:   **function** $\text{FIND}(S, Z, \tau)$        $\triangleright$ Find alteration values
7:      $\texttt{int} \leftarrow \emptyset$        $\triangleright$ Store valid intervals
8:      **for** $\langle G, \boldsymbol{\theta}, \boldsymbol{\varepsilon} \rangle \in S$ **do**
9:        $[u, v] \leftarrow$ solution of $\mathbf{M}(\mathbf{A}\mathbf{x} + \mathbf{B}\mathbf{z}^{\xi} + \mathbf{C}\boldsymbol{\varepsilon}) \leq \mathbf{d}$   $\triangleright$ $\mathbf{A}, \mathbf{B}, \mathbf{C}$ are determined by $G, \boldsymbol{\theta}, Z$
10:        $\texttt{int} \leftarrow \texttt{int} \cup \{[u, v] \cap \Delta(Z)\}$
11:      **return** $\text{FILTERINTERVALS}(\texttt{int}, \lceil n\tau \rceil)$
12:   **function** $\text{FILTERINTERVALS}(\mathcal{I}, k)$        $\triangleright$ Find elements that appear more than $k$ times in $\mathcal{I}$
13:      $\texttt{arr} \leftarrow \emptyset$
14:      **for** $[u, v] \in \mathcal{I}$ **do**
15:        $\texttt{arr} \leftarrow \texttt{arr} \cup \{(u, 0)\}$
16:        $\texttt{arr} \leftarrow \texttt{arr} \cup \{(v, 1)\}$        $\triangleright$ 0 or 1 indicates the value is taken from the left or the right endpoints of an interval
17:      $\texttt{arr} \leftarrow \text{sorted}(\texttt{arr})$        $\triangleright$ Sorted in increasing order with the first element as the key
18:      $\texttt{cnt} \leftarrow 0$
19:      **for** $0 \leq \texttt{i} < |\texttt{arr}|$ **do**
20:        **if** $\texttt{arr}[\texttt{i}][1] = 0$ **then**        $\triangleright$ If the value is a left endpoint
21:          $\texttt{cnt} \leftarrow \texttt{cnt} + 1$
22:        **else**
23:          $\texttt{cnt} \leftarrow \texttt{cnt} - 1$
24:        $\texttt{arr}[\texttt{i}][1] \leftarrow \texttt{cnt}$        $\triangleright$ Record the number of intervals that intersect at $\texttt{arr}[\texttt{i}][0]$
25:      $\texttt{rtn} \leftarrow \emptyset$
26:      $\texttt{left} \leftarrow \texttt{NULL}, \texttt{right} \leftarrow \texttt{NULL}$
27:      **for** $0 \leq \texttt{i} < |\texttt{arr}|$ **do**        $\triangleright$ Scan the array and record valid intervals
28:        $\texttt{right} \leftarrow \texttt{arr}[\texttt{i}][0]$
29:        **if** $\texttt{arr}[\texttt{i}][1] \geq k$ **then**
30:          **if** $\texttt{left}$ is $\texttt{NULL}$ **then**
31:            $\texttt{left} \leftarrow \texttt{arr}[\texttt{i}][0]$
32:        **else**
33:          **if** $\texttt{left}$ is not $\texttt{NULL}$ **then**
34:            $\texttt{rtn} \leftarrow \texttt{rtn} \cup \{[\texttt{left}, \texttt{right}]\}$
35:            $\texttt{left} \leftarrow \texttt{NULL}$
36:      **return** $\texttt{rtn}$

**Output:** The found alterations $\mathcal{C}$

---

mean and variance at an unseen $\mathbf{z}_{l+1}$ by $\mu_g(\mathbf{z}_{l+1})$ and $\sigma_g^2(\mathbf{z}_{l+1})$ respectively. Given queried points $\{\mathbf{z}^1, \ldots, \mathbf{z}^l\}$, we use upper confidence bound [60] as a querying criterion to select the next point:

$$\mathbf{z}^{l+1} = \underset{\mathbf{z}^{l+1} \in \mathcal{C}(\mathbf{Z}^\xi)}{\arg\max} \quad \mu_g(\mathbf{z}^{l+1}) + \gamma \cdot \sigma_g(\mathbf{z}^{l+1}),$$

where $\gamma$ is a constant and $\mathcal{C}(\mathbf{Z}^\xi)$ is the feasible region of $\mathbf{Z}^\xi$ defined by $\mathcal{C}$. We run this optimization process for every alteration target in $\mathcal{C}$ to find the corresponding best alteration values. Finally, the alteration, including both target variables and values, that has maximum mutual information is selected as the recommended output.

---

**Algorithm 4** Mutual information estimation

---

**Input:** The posterior estimate $p(G, \boldsymbol{\theta} \mid D_t)$, the candidate graph set $\mathcal{G}$, the observed $\mathbf{x}$, the alteration $\xi$, the number of samples for estimating an entropy term $n$, the number of samples for structural parameters $n'$.
1: $srms \leftarrow \emptyset$
2: **for** $G \in \mathcal{G}$ **do**
3:   **for** $i \in [\lceil n \cdot p(G \mid D_t) \rceil]$ **do**
4:    Sample $\boldsymbol{\theta}$ from $p(\boldsymbol{\theta} \mid G, D_t)$
5:    $srms \leftarrow srms \cup \{\langle G, \boldsymbol{\theta}_i \rangle\}$
6: $samples \leftarrow \emptyset$
7: **for** $\langle G, \boldsymbol{\theta} \rangle \in srms$ **do**
8:   Sample $\mathbf{z}$ and $\mathbf{y}$ from $p(\mathbf{z}, \mathbf{y} \mid G, \boldsymbol{\theta}, \mathbf{x}, \xi)$
9:   $samples \leftarrow samples \cup \{(\mathbf{z}, \mathbf{y})\}$
10: $\hat{H}_1 \leftarrow \text{ENTROPYESTIMATE}(samples)$
11: $\hat{H}_2 \leftarrow 0$
12: **for** $G \in \mathcal{G}$ **do**
13:   $entropies \leftarrow \emptyset$
14:   **for** $i \in [n']$ **do**
15:    Sample $\boldsymbol{\theta}$ from $p(\boldsymbol{\theta} \mid G, D_t)$
16:    $samples \leftarrow \emptyset$
17:    **for** $i \in [n]$ **do**
18:     Sample $\mathbf{z}$ and $\mathbf{y}$ from $p(\mathbf{z}, \mathbf{y} \mid G, \boldsymbol{\theta}, \mathbf{x}, \xi)$
19:     $samples \leftarrow samples \cup \{(\mathbf{z}, \mathbf{y})\}$
20:    $entropies \leftarrow entropies \cup \{\text{ENTROPYESTIMATE}(samples)\}$
21:   $cond\_entropy \leftarrow$ average of $entropies$
22:   $\hat{H}_2 \leftarrow \hat{H}_2 + cond\_entropy \cdot p(G \mid D_t)$
23: $\hat{I} \leftarrow \hat{H}_1 - \hat{H}_2$
**Output:** The mutual information estimate $\hat{I}$

---

# D Proofs

In this section, we provide proof for claims in the main text.

**Proposition 4.** *For any $\langle G_1, \boldsymbol{\theta}_1 \rangle$ and $\langle G_2, \boldsymbol{\theta}_2 \rangle$ on $\mathbf{X}_t \cup \mathbf{Z}_t \cup \mathbf{Y}_t$, if they differ only in describing the generation of $\mathbf{X}_t$, then for any $\mathbf{X}_t = \mathbf{x}_t$ and alterations $\xi$ on $\mathbf{Z}_t$, we have*

$$\mathbb{P}\left(\mathbf{y}_t \mid \mathbf{x}_t, Rh(\xi); G_1, \boldsymbol{\theta}_1\right) = \mathbb{P}\left(\mathbf{y}_t \mid \mathbf{x}_t, Rh(\xi); G_2, \boldsymbol{\theta}_2\right).$$

*Proof.* We have

$$\mathbb{P}(\mathbf{y} \mid \mathbf{x}, Rh(\xi); G_i, \boldsymbol{\theta}_i)$$
$$= \sum_{\mathbf{z}} \mathbb{P}(\mathbf{z} \mid \mathbf{x}, Rh(\xi); G_i, \boldsymbol{\theta}_i) \mathbb{P}(\mathbf{y} \mid \mathbf{z}, \mathbf{x}, Rh(\xi); G_i, \boldsymbol{\theta}_i), \quad i = 1, 2.$$

Since $(G_1, \boldsymbol{\theta}_1)$ and $(G_2, \boldsymbol{\theta}_2)$ only differ in parameters controlling $\mathbb{P}(\mathbf{x})$, we have

$$\mathbb{P}(\mathbf{z} \mid \mathbf{x}, Rh(\xi); G_1, \boldsymbol{\theta}_1) = \mathbb{P}(\mathbf{z} \mid \mathbf{x}, Rh(\xi); G_2, \boldsymbol{\theta}_2),$$

and
$$\mathbb{P}(\mathbf{y} \mid \mathbf{z}, \mathbf{x}, Rh(\xi); G_1, \boldsymbol{\theta}_1) = \mathbb{P}(\mathbf{y} \mid \mathbf{z}, \mathbf{x}, Rh(\xi); G_2, \boldsymbol{\theta}_2).$$

Combining the above equations gives
$$\mathbb{P}\left(\mathbf{y} \mid \mathbf{x}, Rh(\xi); G_1, \boldsymbol{\theta}_1\right) = \mathbb{P}\left(\mathbf{y} \mid \mathbf{x}, Rh(\xi); G_2, \boldsymbol{\theta}_2\right).$$

$\square$

**Proposition 5.** *Let $\xi = \{(Z_{a_i}, z_{a_i})\}_{i=1}^k$ be a valid alteration where $Z_{a_i} \in \mathbf{Z}_t$ and $z_{a_i} \in \Delta(Z_{a_i})$. Given $\langle G, \boldsymbol{\theta}, \boldsymbol{\varepsilon} \rangle$ and $\mathbf{x}_t$, the outcome variables are linear in the alteration values. We have*

$$\mathbf{Y}_t = \mathbf{A}\mathbf{x}_t + \mathbf{B}\mathbf{z}^\xi + \mathbf{C}\boldsymbol{\varepsilon}, \tag{8}$$

*where $\mathbf{z}^\xi = (z_{a_1}, \dots, z_{a_k})^T$. $\mathbf{A}$, $\mathbf{B}$, and $\mathbf{C}$ are constant matrices of appropriate shapes and are uniquely determined by $G$, $\boldsymbol{\theta}$ and altered variables $\mathbf{Z}^\xi = \{Z_{a_i}\}_{i=1}^k$.*

*Proof.* The proposition is immediate by noting that if the parent of a set of variables can be described by
$$\mathbf{PA} = \mathbf{A}'\mathbf{x} + \mathbf{B}'\mathbf{z}^\xi + \mathbf{C}'\boldsymbol{\varepsilon},$$

then
$$\mathbf{V} = \boldsymbol{\beta}^T \mathbf{PA} + \varepsilon_{\mathbf{V}} = \boldsymbol{\beta}^T \mathbf{A}'\mathbf{x} + \boldsymbol{\beta}^T \mathbf{B}'\mathbf{z}^\xi + \boldsymbol{\beta}^T \mathbf{C}'\boldsymbol{\varepsilon} + \varepsilon_{\mathbf{V}}$$

can be described in a similar linear form. $\square$

**Theorem 6.** *Unless $\mathrm{P} = \mathrm{NP}$, finding a $k$-dimensional $\mathbf{z}^\xi \in \Delta(\mathbf{Z}^\xi)$ that satisfies $\sum_{i=1}^n \mathbb{I}(\mathbf{M}(\mathbf{A}_i\mathbf{x} + \mathbf{B}_i\mathbf{z}^\xi + \mathbf{C}_i\boldsymbol{\varepsilon}_i) \leq \mathbf{d}) \geq n \cdot \tau$ (if there is a valid solution) is not solvable with any algorithm of running time polynomial in $k$ and $n$.*

*Proof.* We prove this by contradiction. Assume that the problem is solvable with a subroutine $f$ that runs in polynomial time. Then, given any infeasible linear system $\Sigma : \{\mathbf{Ax} \leq \mathbf{b}\}$, we can find a feasible subsystem containing as many inequalities as possible by finding a maximized $\tau$ that gives a valid solution for the problem of finding a $\tau$ that satisfies at least $n \cdot \tau$ inequalities. And the process of finding a maximized $\tau$ can be finished by running a binary search on $\tau$ and calling $f$, which will be called at most $O(\log n)$ times. Thus the overall running time will still be polynomial. However, finding a maximum feasible subsystem is known to be NP-hard [21], which cannot be solved within polynomial time if $\mathrm{P} \neq \mathrm{NP}$. Therefore we have a contradiction and conclude that the problem does not admit a polynomial time algorithm. $\square$

**Theorem 7.** *Given an observed evidence $\mathbf{x}_t$, an alteration $\xi$, and a desired set $\mathcal{S} = \{\mathbf{y} \in \mathbb{R}^{|\mathbf{Y}|} \mid \mathbf{My} \leq \mathbf{d}\}$, let $S_{eval} = \{\langle G_i, \boldsymbol{\theta}_i, \boldsymbol{\varepsilon}_i \rangle\}_{i=1}^n$ be $n$ i.i.d. samples from $\mathbb{P}(G, \boldsymbol{\theta}, \boldsymbol{\varepsilon} \mid D_t)$. Let $\{\mathbf{y}^i\}_{i=1}^n$ be generated from $Rh(\xi)$ and $S_{eval}$ following the structural equations in Eq. (1). Define $n_o \triangleq |\{i \mid \mathbf{y}^i \notin \mathcal{S}\}|$. Let $\hat{p} \triangleq 1 - n_o/n$, $\underline{p} \triangleq 1 - F^{-1}(1 - \delta/(2n); n_o + 1, n - n_o)$ if $n_o < n$ otherwise $\underline{p} \triangleq 0$, and $\bar{p} \triangleq 1 - F^{-1}(\delta/(2n); n_o, n - n_o + 1)$ if $n_o > 0$ otherwise $\bar{p} \triangleq 1$, where $F^{-1}(\cdot; \alpha, \beta)$ is the inverse cumulative distribution function of the beta distribution with parameters $\alpha$ and $\beta$. For any $\delta \in (0, 1)$, with probability at least $1 - \delta$, we have*

$$\max\left\{\underline{p}, \hat{p} - \sqrt{\ln(2/\delta)/2n}\right\} \leq \mathbb{P}(\mathbf{Y}_t \in \mathcal{S} \mid D_t, \mathbf{x}_t, Rh(\xi)) \leq \min\left\{\bar{p}, \hat{p} + \sqrt{\ln(2/\delta)/2n}\right\}.$$

*Proof.* The proof is mainly based on the scenario approach [23] and adapted from Badings et al. [24]. The desired region for $\mathbf{Y}$ is given by $\mathcal{S} = \{\mathbf{y} \in \mathbb{R}^{|\mathbf{Y}|} \mid \mathbf{My} \leq \mathbf{d}\}$. We define a scaled version of $\mathcal{S}$ by

$$R(\lambda) = \left\{\mathbf{y} \in \mathbb{R}^{|\mathbf{Y}|} \mid \mathbf{My} \leq \lambda\mathbf{d} + (1 - \lambda)\mathbf{Mh}\right\}, \tag{19}$$

where $h \in \mathbb{R}$ is a Chebyshev center of $\mathcal{S}$ [61]. We have $R(1) = \mathcal{S}$ and $R(\lambda_1) \subset R(\lambda_2)$ for any $0 \leq \lambda_1 < \lambda_2$. Using Prop. 5, we express each $\mathbf{y}^i$ with

$$\mathbf{y}^i = \mathbf{A}_i\mathbf{x} + \mathbf{B}_i\mathbf{z}^\xi + \mathbf{C}_i\boldsymbol{\varepsilon}_i. \tag{20}$$

We use the scenario approach to estimate the probability of the happening of undesired $\mathbf{Y}$ with the following convex scenario optimization problem with discarded samples:

$$\min_{\lambda \geq 0} \quad \lambda$$
$$\text{s.t.} \quad \mathbf{y}^i \in R(\lambda) \ \forall i \in \{1, \dots, n\} \setminus Q, \tag{21}$$

which has a scalar decision variable $\lambda$. We denote the optimal solution of the above optimization program with $\lambda^*_{|Q|}$. $Q$ is a subset of samples whose constraints have been discarded according to the following rule [24, Lem. 1]: The sample removal set $Q \subseteq \{1, \dots, n\}$ is obtained by iteratively removing the active constraints from (21). Thus, given $N$ samples and any two removal sets with cardinalities $|Q_1| < |Q_2|$, it holds that $Q_1 \subset Q_2$. Moreover, any discarded sample $i \in Q$ violates the solution $\lambda^*_Q$ to Eq. (21), *i.e.* $\mathbf{y}^i \notin R_j\left(\lambda^*_{|Q|}\right)$, with probability one. When discarding $|Q| = n_o$ samples, it holds that $R(\lambda^*_{n_o}) \subseteq \mathcal{S}$. When discarding $|Q| = n_o - 1$ samples $(n_o > 0)$, $R(\lambda^*_{n_o-1}) \supset \mathcal{S}$.

We further assume that given a sample $\langle G_i, \boldsymbol{\theta}_i, \boldsymbol{\varepsilon}_i \rangle$ from $\mathbb{P}(G, \boldsymbol{\theta}, \boldsymbol{\varepsilon} \mid D_t)$, the probability that the generated $\mathbf{y}^i$ is on the boundary of any polytope $R(\lambda)$ for any $\lambda \geq 0$ is zero.

Based on the results of Romao et al. [62, Thm. 5] and Badings et al. [24, Thm. 1], we have

$$\mathbb{P}\left\{ P\left(\mathbf{y} \notin R(\lambda^*_{|Q|})\right) \leq \epsilon \right\} = F(\epsilon; |Q| + 1, n - |Q|), \tag{22}$$

where $F(\cdot; \alpha, \beta)$ is the cumulative distribution function of the beta distribution with parameters $\alpha$ and $\beta$, $|Q| < n$. Equivalently, we have

$$\mathbb{P}\left\{ P\left(\mathbf{y} \notin R(\lambda^*_{|Q|})\right) > \epsilon \right\} = 1 - F(\epsilon; |Q| + 1, n - |Q|). \tag{23}$$

Solving $1 - F(\epsilon; |Q| + 1, n - |Q|) = \frac{\delta}{2n}$, we have

$$\epsilon = F^{-1}(1 - \frac{\delta}{2n}; |Q| + 1, n - |Q|). \tag{24}$$

For notational convenience, let $F_i(\cdot) = F(\cdot; i + 1, n - i)$. We have

$$\mathbb{P}\left\{ P\left(\mathbf{y} \notin R(\lambda^*_{|Q|})\right) > F^{-1}_{|Q|}(1 - \frac{\delta}{2n}) \right\} = \frac{\delta}{2n}. \tag{25}$$

Using Boole's inequality, we know that

$$\mathbb{P}\left\{ \bigcup_{i=0}^{n-1} P\left(\mathbf{y} \notin R(\lambda^*_i)\right) > F^{-1}_i(1 - \frac{\delta}{2n}) \right\} \leq \sum_{i=0}^{n-1} \mathbb{P}\left\{ P\left(\mathbf{y} \notin R(\lambda^*_i)\right) > F^{-1}_i(1 - \frac{\delta}{2n}) \right\} = \frac{\delta}{2}. \tag{26}$$

Therefore, we have

$$\mathbb{P}\left\{ \bigcap_{i=0}^{n-1} P\left(\mathbf{y} \notin R(\lambda^*_i)\right) \leq F^{-1}_i(1 - \frac{\delta}{2n}) \right\} \geq 1 - \frac{\delta}{2}. \tag{27}$$

After observing $\mathbf{y}^i$s at hand, we replace $|Q|$ by $n_o$. Since $R(\lambda^*_{n_o}) \subset \mathcal{S}$ and $\bigcap_{i=0}^{n-1} P\left(\mathbf{y} \notin R(\lambda^*_i)\right) \leq F^{-1}_i(1 - \frac{\delta}{2n})$ implies $P\left(\mathbf{y} \notin R(\lambda^*_{n_o})\right) \leq F^{-1}_{n_o}(1 - \frac{\delta}{2n})$, we have

$$\mathbb{P}\left\{ P\left(\mathbf{y} \notin \mathcal{S}\right) \leq F^{-1}_{n_o}(1 - \frac{\delta}{2n}) \right\} \geq \mathbb{P}\left\{ P\left(\mathbf{y} \notin R(\lambda^*_{n_o})\right) \leq F^{-1}_{n_o}(1 - \frac{\delta}{2n}) \right\} \geq 1 - \frac{\delta}{2}, \tag{28}$$

which further gives

$$\mathbb{P}\left\{ P\left(\mathbf{y} \in \mathcal{S}\right) \geq 1 - F^{-1}_{n_o}(1 - \frac{\delta}{2n}) \right\} \geq 1 - \frac{\delta}{2}. \tag{29}$$

When $n_o = n$, $P\left(\mathbf{y} \in \mathcal{S}\right) \geq 0$ trivially holds. Thus, $\underline{p} \leq \mathbb{P}\left(\mathbf{Y}_t \in \mathcal{S} \mid D_t, \mathbf{x}_t, Rh(\xi)\right)$ holds with probability at least $1 - \frac{\delta}{2}$. For the other half of the lower bound, we apply Hoeffding's inequality on binary variables $D_i \triangleq \mathbb{I}(\mathbf{y}^i \in \mathcal{S})$. Note that $\sum_{i=1}^{n} D_i = n_o$ and $\mathbb{E}(D_i) = \mathbb{P}\left(\mathbf{Y}_t \in \mathcal{S} \mid D_t, \mathbf{x}_t, Rh(\xi)\right)$, we have

$$\mathbb{P}\left\{ n_o - n \cdot \mathbb{P}\left(\mathbf{Y}_t \in \mathcal{S} \mid D_t, \mathbf{x}_t, Rh(\xi)\right) \geq \sqrt{\frac{n}{2} \ln \frac{2}{\delta}} \right\} \leq \frac{\delta}{2}, \tag{30}$$

that is,

$$\mathbb{P}\left\{\mathbb{P}\left(\mathbf{Y}_t \in \mathcal{S} \mid D_t, \mathbf{x}_t, Rh(\xi)\right) \geq \hat{p} - \sqrt{\frac{\ln(2/\delta)}{2n}}\right\} \geq 1 - \frac{\delta}{2}. \tag{31}$$

Combining (29) and (31), we have with probability at least $1 - \frac{\delta}{2}$,

$$\max\left\{\underline{p}, \hat{p} - \sqrt{\frac{\ln(2/\delta)}{2n}}\right\} \leq \mathbb{P}\left(\mathbf{Y}_t \in \mathcal{S} \mid D_t, \mathbf{x}_t, Rh(\xi)\right). \tag{32}$$

For the upper bound, we adopt a similar approach. From Eq. (22), by noting that $P\left(\mathbf{y} \notin R(\lambda^*_{|Q|})\right) + P\left(\mathbf{y} \in R(\lambda^*_{|Q|})\right) = 1$, we get

$$\mathbb{P}\left\{P\left(\mathbf{y} \in R(\lambda^*_{|Q|})\right) \geq 1 - \epsilon\right\} = F_{|Q|}(\epsilon). \tag{33}$$

Substituting $F_{|Q|}(\epsilon)$ with $\frac{\delta}{2n}$, we have

$$\mathbb{P}\left\{P\left(\mathbf{y} \in R(\lambda^*_{|Q|})\right) \geq 1 - F^{-1}_{|Q|}(\frac{\delta}{2n})\right\} = \frac{\delta}{2n}. \tag{34}$$

Again via Boole's inequality,

$$\mathbb{P}\left\{\bigcap_{i=0}^{n-1} P\left(\mathbf{y} \in R(\lambda^*_i)\right) \leq 1 - F^{-1}_i(\frac{\delta}{2n})\right\} \geq 1 - \frac{\delta}{2}. \tag{35}$$

When $n_o > 0$, since $R(\lambda^*_{n_o-1}) \supset \mathcal{S}$ and $\bigcap_{i=0}^{n-1} P\left(\mathbf{y} \in R(\lambda^*_i)\right) \leq 1 - F^{-1}_i(\frac{\delta}{2n})$ implies $P(\mathbf{y} \in R(\lambda^*_{n_o-1})) \leq 1 - F^{-1}_{n_o-1}(\frac{\delta}{2n})$, we have

$$\mathbb{P}\left\{P\left(\mathbf{y} \in \mathcal{S}\right) \leq 1 - F^{-1}_{n_o-1}(\frac{\delta}{2n})\right\} \geq \mathbb{P}\left\{P\left(\mathbf{y} \in R(\lambda^*_{n_o-1})\right) \leq 1 - F^{-1}_{n_o-1}(\frac{\delta}{2n})\right\} \geq 1 - \frac{\delta}{2}. \tag{36}$$

When $n_o = 0$, $P\left(\mathbf{y} \in \mathcal{S}\right) \leq 1$ trivially holds. Thus,

$$\mathbb{P}\left(\mathbf{Y}_t \in \mathcal{S} \mid D_t, \mathbf{x}_t, Rh(\xi)\right) \leq \bar{p} \tag{37}$$

holds with probability at least $1 - \frac{\delta}{2}$. Applying Hoeffding's inequality from the other direction of Eq. (30), we get

$$\mathbb{P}\left\{\mathbb{P}\left(\mathbf{Y}_t \in \mathcal{S} \mid D_t, \mathbf{x}_t, Rh(\xi)\right) \leq \hat{p} + \sqrt{\frac{\ln(2/\delta)}{2n}}\right\} \geq 1 - \frac{\delta}{2}. \tag{38}$$

Combining the above results gives the upper bound that holds with probability at least $1 - \frac{\delta}{2}$:

$$\mathbb{P}\left(\mathbf{Y}_t \in \mathcal{S} \mid D_t, \mathbf{x}_t, Rh(\xi)\right) \leq \min\left\{\bar{p}, \hat{p} + \sqrt{\frac{\ln(2/\delta)}{2n}}\right\}. \tag{39}$$

Combining the lower bound (32) and the upper bound (39) gives the overall bound. $\qquad\square$

# E  Data Details

## E.1  Ride-Hailing Data

We give details about the Ride-Hailing data in this section. The variables included in the generation process are

- weather: weather condition;
- #user: number of users in the neighborhood;

- recommend: recommendation level of the ride-hailing app for a specific route;
- congestion: traffic congestion on the route;
- time: time spent for the ride;
- discount: discount provided by the app;
- rating: user rating for the ride.

The rehearsal graph for the variables is given in Fig. 6. The presumed actionable variables that can be altered by the app are 'recommend' and 'discount'.

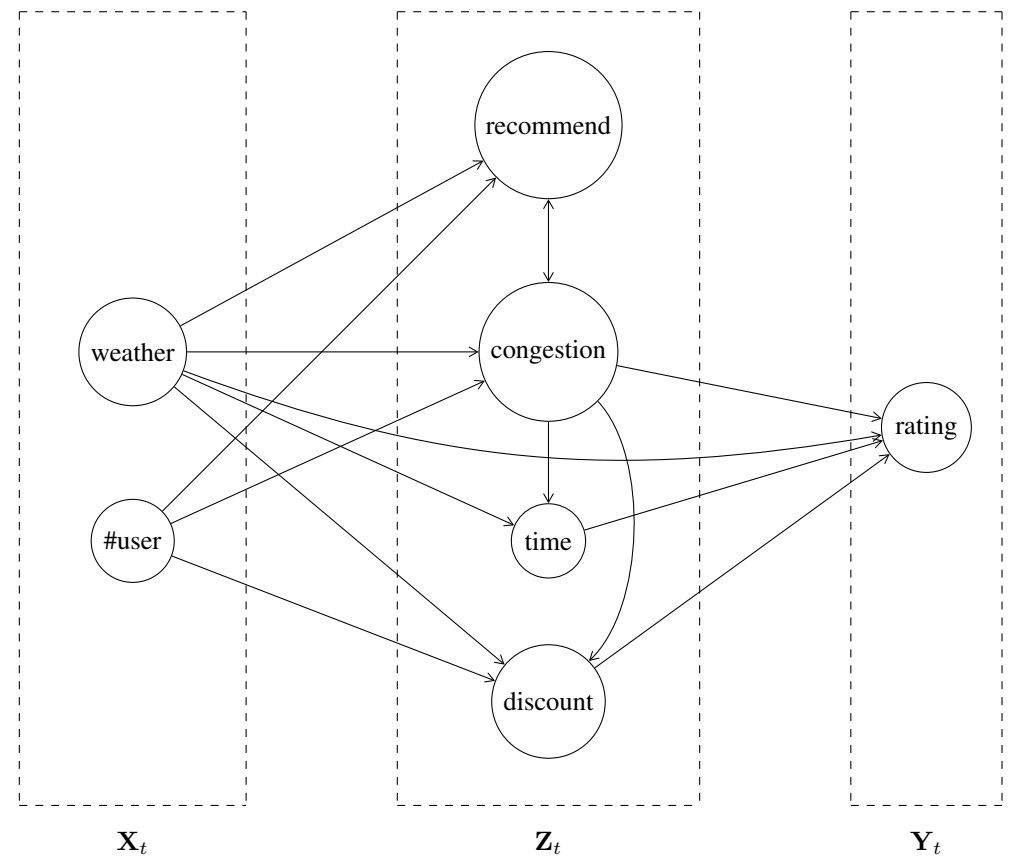

Figure 6: The rehearsal graph for ride-hailing data.

## E.2 Bermuda Data

We give details about the Bermuda data in this section. The Bermuda data involves a set of environmental variables [57, 58, 37]. The variables included in the generation process are

- Chla: sea surface chlorophyll a;
- Sal: sea surface salinity;
- TA: seawater total alkalinity;
- DIC: seawater dissolved inorganic carbon;
- $CO_2$: seawater $P_{CO_2}$;
- Temp: bottom temperature;
- NEC: net ecosystem calcification;
- Light: bottom light levels;
- Nut: PC1 of $NH_4$, $NiO_2 + NiO_3$, $SiO_4$;

- pHsw: seawater pH
- $\Omega_A$: seawater saturation with respect to aragonite.

The rehearsal graph for the variables is given in Fig. 7. The presumed actionable variables that can be altered are: DIC, TA, $\Omega_A$, Chla, and Nut.

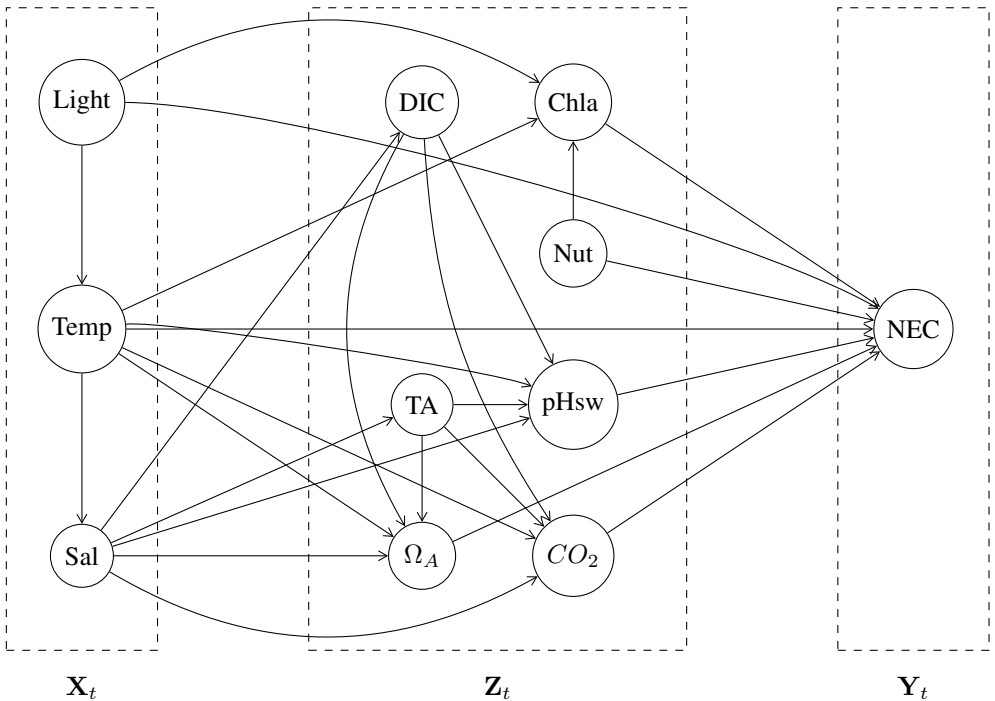

Figure 7: The rehearsal graph for Bermuda data.

