# OpenReview forum: "Rehearsal Learning for Avoiding Undesired Future"
_NeurIPS.cc/2023/Conference — NeurIPS 2023 poster_

### Official Review · Reviewer_Pcym · 2023-07-04

**Soundness:** 2 fair
**Presentation:** 3 good
**Contribution:** 2 fair
**Rating:** 6
**Confidence:** 2

**Summary:**

The paper proposes the framework of Structural Rehearsal Models (SRMs) which is similar to Structural Causal Models but not based on causal relations but instead on rehearsation [1]. In this kind of relation, edges can be bi-directional and indicate that some values influence each other. SRMs are then used to tackle the problem of Avoiding Undesired Future (AUF), where the agent wants to manipulate some intermediate variables Z in order to keep the value of final variable Y within the desired range; the problem is formulated using SRMs. Then, the algorithm is proposed for the case with linear dependencies and one-node interventions. Experiments in toy environments are conducted, showing that the proposed algorithm outperforms off-the-shelf RL algorithms.

[1] https://link.springer.com/article/10.1007/s11704-022-2900-0


**Strengths:**

[S1] The ideas presented in the paper are novel and seem to be well motivated by the need of having a less strict variant of “causal”-based models, where we still can leverage the structure without requiring strict causality.

[S2] The paper is nicely written and well structured, mostly easy to follow.


**Weaknesses:**

[W1] I think having a concrete case where you show that your algorithm works well and causality-based methods either cannot be used or work worse is crucial. I am not 100% convinced by the experiments in the paper. Can causality-based methods be used there (if needed, by lumping together the variables from one clique)? If so, some causality-based methods should be included in the comparison. If not, please provide a compelling explanation why this is not feasible.

[W2] I was not able to understand how the computation graph is changed when the intervention changes size of some clique; I present more details in [Q1].


**Questions:**

[Q1] What happens if you have some clique, e.g. with 3 elements, and then one of the elements is used in the intervention? What will be the function to produce values for the remaining 2 elements, and how does it relate to an un-intervened function? Is it derived from the base one, or are there completely new parameters to learn here? In the latter case, there would be 2^c possible parameter sets for each clique of size c, right?

[Q2] Section 4.2 - can you please explain how you derive the initial set of graphs G?

[Q3] You mention in the paper that your framework enables non-stationary graphs etc. But this is never actually used, and experiments are for the simpler case. Moreover, it seems that this time-dependence could be introduced also for normal SCMs and it is actually orthogonal to the question if you use SCMs or SRMs. Can you maybe elaborate more on that aspect and argue if it is the case that this non-stationarity is more natural in the case of SRMs?


**Limitations:**

I suggest to the authors to include a subsection or a paragraph dedicated to listing the limitations, e.g.: focusing on linear case, toy environments, etc.

---

> ### Author Rebuttal · Authors · 2023-08-06
>
> Thanks for your insightful comments! Below we address your questions in a point-by-point fashion.
>
> > W1. Regarding the experiments and if causality-based methods can be used (if needed, by lumping together the variables from one clique).
>
> Thanks for the insightful questions! We want to emphasize that the primary goal of the experiments was not to demonstrate that our method outperforms specific baselines. Instead, as the first attempt to explore rehearsal learning, we aimed to illustrate through the experiments that building decision-making models based on rehearsal relations (rather than relying on correlation or causation) is feasible and sensible, at least for certain problems. The experiments clearly achieved this objective.
>
> Regarding the applicability of causality-based methods by lumping together the variables from one clique, we are afraid that it is not a suitable approach. While lumping together variables might seem like an intuitive way to incorporate causality, it results in coarse modeling granularity, making it challenging to find appropriate decisions. For example, if we lump three interrelated variables $V_1,V_2,V_3$ together as a single node $W$ in a causal graph, then the atomic intervention would be a joint intervention on $W=(V_1,V_2,V_3)$ simultaneously. The joint intervention on the three variables can be infeasible or unnecessarily costly for some decision tasks. Furthermore, in this lumping approach, we cannot model size-1 interventions in the "causal" graph by selectively intervening on only one variable, let's say $V_1$, while keeping the other two, $V_2$ and $V_3$, untouched. The reason is that intervening on $V_1$ would result in changes in $V_2$ and $V_3$, given their interrelated nature. In less formal terms, that means performing $do(W=w)$, which actually intervenes on an individual component, could lead to $W\ne w$, which does not make sense in causality. Consequently, this approach is not well-suited for capturing the intricate relationships among interrelated variables. On the other hand, using SRM to model these variables are more appropriate, enabling us to account for fine-grained interventions and accurately capture their effects on the system.
>
> > W2 and Q1. Regarding the change of the graph when the intervention changes the size of some clique.
>
> Consider an example with a clique of size three. When intervening on one variable, the generating mechanism for the other two variables uses another set of parameters, which is generally different from those in generating the un-intervened function and is not derived from the base one if no further assumptions are made. Hence, in the most general case, there could be $2^c$ possible parameter sets for each clique of size $c$, as you correctly pointed out. In practical implementations, it is reasonable to limit the size of a clique and maintain the number of parameter sets to an acceptable level. Moreover, it is unlikely that we would need to or be able to intervene on every possible subset of a clique. We can save the parameters corresponding to those generating mechanisms that would not be used, thus reducing unnecessary computational burden. Also, we can impose certain assumptions on the relations among the parameters. For instance, it is possible to marginalize the parameters of the base case and use them as priors for the other sets of parameters, leading to more efficient modeling.
>
> > Q2. Regarding the generation of the initial set of graphs.
>
> The generation of the initial set of graphs $\mathcal{G}$ is given in Section 4.1, where we adopt a bootstrapped-based method. Specifically, we obtain a candidate graph $G$ by sampling from the observed data with replacement and applying a graph learning method on the re-sampled data, such as the preliminary learning method introduced in Appendix B. $\mathcal{G}$ is constructed by repeating this procedure $|\mathcal{G}|$ times. Each graph $G \in \mathcal{G}$ is given equal weight to serve as the prior distribution of the graphs. Subsequently, as we receive examples in each round, we update the posterior distribution of these graphs accordingly. We can then sample graphs from this posterior distribution in the following steps.
>
> > Q3. Regarding the time-dependence of SRM.
>
> This paper primarily focuses on demonstrating the feasibility of using rehearsal to enhance decision-making in a basic setting. Non-stationary environment modeling is a more general and practical consideration but is also a much more complicated topic that requires further research and investigation. Intuitively, it is possible to model environments at time $t$ with a different causal graph and describe them with multiple SCMs. But the temporal nature is not inherent in SCMs. And current temporal causal modeling generally assumes that the causal structure in a dynamic system simply repeats over time, which is different from the situation we considered where the relations among variables can evolve over time. As for SRMs, since it is designed for modeling decision-making tasks, where time is an indispensable component, it is natural for them to have intrinsic time dependence.
>
> Moreover, an interesting observation that may warrant further exploration in future studies is that the interrelated variables in SRMs can provide hints for the possible evolution of variable relations. In an evolving dynamic environment, the change from $A\rightarrow B$ to $A\leftarrow B$ may undergo an intermediate stage where the relation is $A\leftrightarrow B$, which is not considered in causal modeling. From this perspective, SRMs seem to be a more natural choice for describing such evolutions and incorporating structural knowledge into decision-making tasks.
>
> Thank you again for your insightful feedback! We hope that the explanations address your concerns appropriately. We will also incorporate the above discussions into the revised version of the paper.

---

> > ### Comment · Reviewer_Pcym · 2023-08-14
> >
> > Thank you for addressing my questions. I have decided to raise my score to 6.

---

### Official Review · Reviewer_8gB7 · 2023-07-06

**Soundness:** 2 fair
**Presentation:** 3 good
**Contribution:** 2 fair
**Rating:** 5
**Confidence:** 2

**Summary:**

This work presented a rehearsal learning framework to avoid undesired future. The framework was characterized by a
probabilistic graphical model called rehearsal graphs and structual equaitons, and the actionable decisions that enable the
outcome to be altered are found under a bayesian famework, and the correct bound to quantify the associated risk was
offered.

**Strengths:**

1. The problem that the paper is trying to solve is important, how to exploit the data generating mechanism to improve decision-making efficiency is of great importance but not yet fully addressed.

2. The paper is generally well written and clearly motivated.



**Weaknesses:**

I think the biggest problem of this paper is the difference between the Strcutural Rehearsal Model (SRM) and Structural Causal Model (SCM). Authors claimed that SCM cannot address the dynamic problem with time involved which is not true.
There are a large number of existing work which investigate the causal structure learning / inference problem for dynamic systems and they can be leveraged for temporal decision making. The most widely used model is called Granger Causality Model and its variants. I don't see essential difference between SRM and SCM, if the only difference is the so called dynamic issue claimed in the paper.




**Questions:**

1. I think a basic computational costs should be evaluated for the proposed algorithm.
The experiments should add more comparison between the proposed method and existing methods.

2. Could authors please include temporal causal models as baselines?  I would also suggest that authors should compare the proposed method with causal bandit methods.


**Limitations:**

see the weakness and question sections.

---

> ### Author Rebuttal · Authors · 2023-08-06
>
> Thanks for your valuable comments! Below we address your questions in a point-by-point fashion.
>
> > W1. Regarding the difference between the Strcutural Rehearsal Model (SRM) and Structural Causal Model (SCM), and the dynamic issue in decision problems.
>
> Thanks for raising the question! We are sorry for the possible misunderstanding raised by the word "dynamic". We want to clarify the distinction between the "dynamic" setting considered in the causal discovery and causal inference literature and the "dynamic" setting we refer to in this work. In the causal discovery and inference literature, the "dynamic" setting typically involves dynamic systems that expand over time. Related methods commonly assume that the causal structures among variables at consecutive time fragments remain unchanged and simply repeat over time, so that valid causal discovery and inference on time-series data are possible. However, in our work, when we mention the "dynamic" and time-dependent nature of decision-making, we mean that the relations among the same group of variables can evolve (or change) at different time segments. For example, offering a discount may attract more customers initially, but as time evolves and an economic recession occurs, the relations can change: offering the same discount may be useless. This evolving aspect is not considered in previous work on causality.  We will modify the description in the revised version of the paper to make it more comprehensive.
>
> Furthermore, in addition to the "evolving" nature of SRM, the fundamental difference between SRM and SCM is the distinct relations they described, namely rehearsation and causation, which we gave a detailed motivation and explanation in Section 2 and Appendix B. Compared to SCM, SRM additionally considers variables that are interrelated, altering each of them would render other variables to change, which can be inappropriate or inconvenient to model with causal relations. Moreover, we want to emphasize that this work is the first exploration into rehearsation and rehearsal learning. This study has already successfully demonstrated the possibility of building decision-making frameworks based on rehearsation, and revealed the potential advantage over other methods for problems with extremely sparse interactions. We believe that this work could open up exciting avenues for meaningful future investigations into rehearsal modeling-based decision-making.
>
> > Q1. Regarding computational costs and comparison with more baselines.
>
> The average running time of the proposed method on two datasets for $T=100$ rounds is 228.4 and 332.6 seconds, respectively. As the running time can vary a lot with different learning components used in the framework, the actual running time may not be very informative. Hence, we give a simplified analysis of the running time complexity in the response to Q1 raised by Reviewer LZtj. The overall time complexity is about $O(|\mathcal{G}|\text{poly}(d)+T(B d^2 + n' |\mathcal{G}| + d n \log n))$, where $d$ denotes the number of variables and $n$ and $n'$ are respectively the sampling sizes in the candidate action finding step and the Bayesian update step. The time complexity reveals that the proposed method scales polynomially with the number of variables.
>
> As for comparisons with more baselines, we conducted new experiments with the two settings in Section 5 using classic reinforcement learning methods SAC and PPO. The average success probabilities are presented in the following table:
>
> |  | SAC | PPO | DDPG | CATS | Ours|
> | --- | --- | --- | --- | --- | --- |
> |Ride-hailing  | .177 | .154 | .173 | .104 | .714 |
> |Bermuda  | .205 | .190 | .230 | .215 | .679 |
>
> As observed, baselines do not achieve desirable performance, which is probably due to the scarcity of interactions in the AUF problem. However, we want to emphasize that the main purpose of the experiments was not to demonstrate that our method outperforms other baselines. Instead, as the first exploration, we want to show with the experiments that it is possible and sensible to build decision-making on rehearsal relations (instead of correlation and causation), at least for some problems. We hope this work could inspire future investigations into rehearsal-based decision-making.
>
> > Q2. Regarding including temporal causal models and causal bandits as baselines.
>
> Thank you for the suggestion! Solving AUF with causal methods requires the method to be capable of optimizing the probability of belonging to a desired region with unknown causal structures. Unfortunately, we explored various temporal causal methods, but none of them can be applied. Regarding causal bandits, most studies assume a known causal graph [1,2,3], which is not available in our setting. We came across a very recent work [4] that does not require knowledge of the causal graph, and we conducted new experiments using this method. The average success probabilities obtained were far from satisfactory, achieving 0.073 and 0.066 on the two datasets, respectively. We believe that the poor performance is mainly attributed to the fact that causal bandits aim to find a universally good action (arm) with an optimal expected reward, while in the AUF problem, the optimal action depends on the observed $X$, and there is no single action that performs well across all circumstances.
>
> Thank you again for your valuable feedback! We hope that the explanations address your concerns appropriately. We will also incorporate some of the above discussions into the revised version of the paper.
>
> [1] Lattimore et al. Causal Bandits: Learning Good Interventions via Causal Inference. NeurIPS, 2016.
>
> [2] Lee and Bareinboim. Structural Causal Bandits: Where to Intervene? NeurIPS, 2018.
>
> [3] Lee and Bareinboim. Structural Causal Bandits with Non-manipulable Variables. AAAI, 2019.
>
> [4] Malek et al. Additive Causal Bandits with Unknown Graph. ICML. 2023.

---

> > ### Comment · Reviewer_8gB7 · 2023-08-18
> >
> > Thank you very much for the clarification. I am willing to increase my score to 5 as authors' clarification partially addresses my concerns.

---

### Official Review · Reviewer_LgnQ · 2023-07-07

**Soundness:** 3 good
**Presentation:** 3 good
**Contribution:** 3 good
**Rating:** 7
**Confidence:** 3

**Summary:**

The authors present a formulation called the rehearsal learning framework to study problems where reasoning about undesirable future outcomes can be leveraged to avoid those undesirable futures---a kind of forward-looking counterfactual reasoning.  The authors additionally show how decisions can be made within this framework using Bayesian methods, and prove some PAC bounds.

**Strengths:**

Originality:  This work represents a fairly novel formulation of a decision problem, building upon prior work in probabilistic graphical models and causal modeling.

Quality/clarity:  This work is fairly clear, though it is also fairly dense.

Significance: This particular formulation seems important, and is underexplored relative to the more traditional scenarios.  I suspect this method will be an influential launch-point for future work on rehearsal learning.

**Weaknesses:**

* The authors could probably spend more time comparing their method to other classical decision problem baselines (e.g., SAC / PPO, etc.).  There's also something of a cottage industry of sample-efficient methods in the RL literature that are probably relevant here, and it's unclear how much this method is buying you relative to these.  In short, it makes it difficult to situate how good this method really is, without a comparison to a more familiar method or problem.  That said, the experiments provided by the authors in their chosen domains are certainly compelling, and do seem to provide strong evidence of the efficacy of the method.  I just worry the authors have drawn a small box around the problem they care about which exactly coincides with where their method wins, and not terribly well overlapping with the space of problems people actually care to solve.

**Questions:**

* On line 30ish, the authors compare and contrast their formalism of AUF with the usual sequential decision paradigm of RL.  I'm not sure I fully buy this distinction, and I'm pretty sure MDPs can capture exactly as much structure as the authors suggest their formalism is capable of capturing (After all, RL has been applied to and solved many difficult games that have long term structure to be modeled, it's just usually the case that any decisions that might lead to negative rewards far in the future are absorbed into some value function after having seen that detrimental effect happen enough times in, e.g., simulation, and not via some explicit counterfactual/graphical reasoning process).  Could the authors clarify these sentences, or perhaps change the emphasis to be more one of sample efficiency, because I'm not sure I fully buy the strength of the current language?

* The formalism of "rehearsation" proposed by the authors starts looking very similar to the process of, e.g., performing rollouts within a simulator to see if some trajectory stays within some feasibility region, or satisfies some kind of safety constraint (especially common in the robotics literature).  Could the authors comment on how they see their work interacting with this line of work?

* Could the authors provide some analysis of how the method's performance fares as the assumptions around low sample efficiency are relaxed?  Table 1 clearly indicates the superiority of the method, but surely DDPG starts winning after the number of interactions with the environment is increased.  Or do I misunderstand the relevant limits?  (Regardless, it's always a great idea to better articulate where the method starts to fail!)

* The authors are selling their method as one that requires fewer interactions with an environment, but also demonstrate pretty strong sensitivity of their method to the hyperparameter $\tau$ (which is obviously problem dependent).  Could the authors comment here?

---

> ### Author Rebuttal · Authors · 2023-08-06
>
> Thanks for your constructive comments! Below we address your concerns and questions in a point-by-point fashion.
> > W1. Regarding comparison with other baselines and sample-efficient methods (e.g. SAC / PPO).
>
> Thanks for the suggestion! We conducted new experiments with SAC and PPO. The average success probabilities are:
>
> |  | SAC | PPO | Ours|
> | --- | --- | --- | --- |
> |Ride-hailing  | .177 | .154 | .714 |
> |Bermuda  | .205 | .190 | .679 |
>
> We see that limited sample sizes pose challenges for baselines to achieve desired performance. However, we want to clarify that the main aim of the experiments was not to demonstrate that our method outperforms baselines. Instead, we want to show that it is possible and sensible to build decision-making on rehearsal relations (instead of correlation and causation), at least for some problems. It is more like a by-product that the proposed formulation has benefits for some problems with scarce data. And we hope this work could inspire future investigations into rehearsal-based decision-making.
> > Q1. Regarding the distinction between rehearsal formalism and RL, and the structures that MDP can capture.
>
> We acknowledge that MDP-based RL methods can be applied to problems like AUF (Lines 30-31). The MDP formalism surely can capture enough information with sufficient modeling granularity. However, MDP may not be the most appropriate language for certain structural information. Let's consider an example with two actionable variables $A$ and $B$, and an outcome $C$ with the structure $A\leftarrow B \leftrightarrow C$. An MDP model needs to specify the transition probabilities between all possible 3-dimensional states and consider all actions $(A,B)=(a,b)$. On the other hand, by utilizing rehearsal graphs, we know that actioning on $A$ does not have any effect on $C$, and thus we can safely consider actions on $B$ only. Both formalisms are capable of capturing all information involved in the decision process, but in some scenarios, the proposed formalism can reveal more direct and helpful information. Furthermore, as you pointed out, the ability to capture structural information is closely related to sample efficiency. If there are sufficiently many samples, the two formalisms should contain a similar amount of information. But if the sample is limited, which is the main setting considered in this work, using the proposed formalism could be more beneficial.
> > Q2. Regarding the connection between rehearsation and rollouts.
>
> The rehearsal operation and rollouts share some similarities: both involve evaluating the effects of actions by activating the actions in some environments. Their difference lies in the underlying rationale. Rollouts are more akin to an empirical test of a policy, though conducted in simulators rather than real environments. While the rehearsal operation is rooted in the rehearsation relation, which is believed to be useful for decision-making and lies between correlation and causation. By performing rehearsal on a fully specified SRM, we can gain a precise understanding of how the effect of actions is propagated through the rehearsal links and what happens if we manually cut off the links. This level of understanding can be highly informative in guiding decision-making for some critical applications. In contrast, rollouts can be less strict: the simulator can even be some neural networks or other black-box simulators, whose inner mechanisms may not be transparent but are enough for evaluating a policy in some problems. Hence, the precise rehearsal operation and the flexible rollouts may complement each other and offer unique advantages in different problems.
> > Q3. Regarding how the performance fares as the assumptions around low sample efficiency are relaxed.
>
> We conducted new experiments with increased $T$. The average success probabilities of DDPG and our method on the Bermuda dataset are presented below:
>
> | T | 100 | 500 | 1000 | 4000 | 7000 | 10000 | 15000 | 20000 |
> | --- | --- | --- | --- | --- | --- | --- | --- | --- |
> | DDPG | .230 | .243 | .255 | .516 | .642 | .688 | .707 | .708 |
> | Ours | .679 | .687 | .693 | .709 | .703  | .707  | .708 | .709
>
> The performance of DDPG increases as $T$ gets larger, and catches up with the proposed method around $T=15000$. The proposed method also benefits from increased samples and maintains success probabilities greater than 0.7 after $T=4000$. The stability of the proposed method with increased samples is easy to understand: the learned SRMs become more accurate with more data, which in turn helps find good decisions.
>
> > Q4. Regarding the sensitivity to $\tau$.
>
> $\tau$ represents the user's desired level of certainty in avoiding undesired future. A smaller $\tau$ indicates that the user is willing to accept a lower success probability. As long as the success probability is close to $\tau$, we should consider the user's requirement as satisfied, and the problem as successfully addressed. From this perspective, the proposed method succeeded with $\tau=0.3, 0.5, 0.7$, and does not show a strong sensitivity. As for cases with $\tau=0.9$, as explained in our response to Q2 raised by Reviewer 9i1V, the drop in performance was due to the specific implementation. With an alternative implementation, the method achieved improved success probabilities of 0.714 and 0.727 on two datasets.
>
> As you pointed out, if the goal is to maximize the success probability instead of matching a fixed preference, the optimal choice of $\tau$ will heavily depend on the specific problem. The user can start from a large $\tau$ and see if the constraints are satisfiable. If unsatisfiable, the decision-maker can lower $\tau$ and restart the process. This flexibility allows the user to fine-tune the decision-making process and adapt it to specific problems.
>
> Thank you again for your valuable feedback! We hope that the explanations address your concern appropriately. We will incorporate the above discussions into the revised paper.

---

> > ### Comment · Reviewer_LgnQ · 2023-08-14
> >
> > I thank the authors tremendously for their detailed response.  I have raised my score to a 7 in response.

---

### Official Review · Reviewer_9i1V · 2023-07-26

**Soundness:** 3 good
**Presentation:** 3 good
**Contribution:** 3 good
**Rating:** 7
**Confidence:** 3

**Summary:**

This paper presents a new graphical architecture, called SRM, whose goal is to be in-between correlational studies and causality models that are SMC. The idea is that identifying variables influenced by decision on other variables is easier than causality learning, and sufficient for decision making. The decision model is supposed to make decision in different time steps, in order to avoid ending up in undesirable states.

To do so, a new family of graphs is introduced, for which are provided learning and inference algorithms, at least in the case where structural equations are linear with Gaussian noise and where inferences are of the AUF (Avoiding Undesirable Future) kind, which consists in maximizing the probability of the outcome being in desirable states over time-steps.

**Strengths:**

* The article provides a generic abstract framework to make decision-making in a dynamic environment, while providing means to make inferences and find adequate decision

* First experiments are provided that demonstrate the performances of the algorithm, that seems to work as expected except for high required success rates, in which cases it breaks down (due to the drop of the success constraint)

**Weaknesses:**

* Comparison with other decision dynamic probabilistic models: the approach mainly deals with the problem of making decisions in a dynamical setting witht he wish to attain given performances. Other frameworks that come to mind when reading the paper are Markov Decision Processes (where the goal would be to maximize the average hitting time of some states) and also Dynamic influence diagrams (or even Dynamic Bayesian Networks with the possibility of interventions as decisions). As those are dynamic tools allowing to solve decision problems, it is unclear why they cannot apply to the current problems and frameworks?

* Effects of hyper-parameters: the complete learning + inference scheme requires a lot of parameters to fix/tune, and it is unclear how much efforts must go into determining those for the algorithms to work. For instance, how does the sampling of graphs as well as the number of graphs in $\mathcal{G}$ affect the efficiency of the algorithms? If $\mathcal{G}$ is small, the posterior could have only a few positive values, and it is not clear wether this is a problem at all? The same can be said for other steps, as those all require some approximations. Not much is given in this respect in the experiments, including in the supplementary materials.

**Questions:**

As this is not my main area of expertise (to say the least), I have only a few questions besides those raised by the mentioned weaknesses:

* At present, it is unclear to me whether the data sets are synthetically generated or not? If so, is there not a bonus to the presented method to use data generated from SRM?

* Is there no way to prevent the observation made in the experiments, i.e., to have a drastic loss of performance (in terms of success probabilities) if the constraint on the sucess probability cannot be satisfied? Cannot we rather have a bi-objective function scaled into one? It seems very extreme to have only a 0.1 probability of success when pre-specifying a wished probability of 0.9.

* From a decision-theoretic perspective, maximizing the probability of being in a state for one output variable can be seen as a very peculiar decision problem, and in practice one often wish to maximize some expected utility whose value may depend on many factors/variables. In particular, previously mentioned models such as MDP or reinforcement learning methods allow to specify such targets. Is the current framework restricted to probabilities of belonging to a state?

**Limitations:**

Limitations are reaonsably discussed.

---

> ### Author Rebuttal · Authors · 2023-08-06
>
> Thanks for your insightful comments! Below we address your questions in a point-by-point fashion.
>
> > W1. It is unclear why other tools cannot apply to the current problems and frameworks.
>
> Thank you for raising this point! Indeed, other decision tools such as MDP can be applied to the AUF problem. We have conducted experiments with the MDP-based reinforcement learning (RL) method DDPG and contextual bandit method CATS in the original version and added two RL methods in the response to W1 raised by Reviewer LgnQ. However, as suggested by the experiments, in problems like AUF where the number of interactions is extremely sparse and limited, traditional modeling tools that do not leverage structural information may suffer from data scarcity, and resorting to causal knowledge could sometimes be inappropriate and excessive. Therefore, the use of rehearsal relations is a more suitable choice. Moreover, we want to emphasize that one of the key contributions of this work is providing evidence that showcases the feasibility of building decision-making models based on rehearsal relations. We believe that this finding can serve as a starting point for promising future investigations into the proposed rehearsal learning framework.
>
> > W2. Regarding the effects of hyper-parameters.
>
> The main hyper-parameters in the proposed framework are the size of $\mathcal{G}$ and the number of samples in each approximation step. Unfortunately, the effect of these parameters is obviously problem-dependent and may not have a universally applicable answer.
>
> To empirically explore the impact of these hyper-parameters, we conducted new experiments using the two settings in Sec. 5. The method exhibited similar performances with the number of samples ranging from 100 to 10,000. But since samples used in approximation are mainly drawn from the posterior distribution, whose computational costs are minor compared to the costs of taking real actions, it is reasonable to set a large sample size. And when doubling the number of graphs (originally set to the size of a learnable equivalence class), we observed that the posterior probability quickly concentrated on a small set of graphs, which had a limited effect on the subsequent rounds. The final average success probabilities for $\tau=0.7$ were 0.707 and 0.682 for the two datasets, respectively, almost matching the previous performance. On the other hand, halving the size resulted in degraded performance (0.577 and 0.474). The rationale behind the results is probably that graphs in a small set $\mathcal{G}$ are likely to be all far from the true graph, leading to poor approximation. In contrast, for a large $\mathcal{G}$, the weights of unreliable graphs diminish after only a few rounds and have little impact on the future learning process. These findings suggest that setting a large $\mathcal{G}$ could be a good choice if there are sufficient computational resources.
>
> >  Q1. Whether the data sets are synthetically generated or not? If so, is there not a bonus to the presented method to use data generated from SRM?
>
> Yes, the datasets were synthetically generated since otherwise we would not be able to evaluate the performance of all methods due to the lack of true environments. The data were generated from SRMs because SRM is an appropriate way of describing the data-generating process in many problems. However, it is worth noting that the data can also be equivalently generated using an MDP formulation with sufficiently many (or continuous) states and actions. From this perspective, we think that the data-generating mechanisms should not be taken as a bonus for the proposed method. Instead, it is an inherent advantage of the proposed method to be able to find and utilize the structural information from data.
>
> > Q2. Is there no way to prevent the drastic loss of performance if the constraint on the success probability cannot be satisfied? Cannot we rather have a bi-objective function scaled into one?
>
> The drastic loss of performance is certainly preventable, and the bi-objective you mentioned is surely a feasible solution. As explained in the last paragraph in Sec. 5, the reason for the drastic loss is that in our implementation, once the method finds the constraint cannot be satisfied, it drops the constraint and just find an action that could maximize the mutual information. We also mentioned an alternative implementation for this situation in Lines 262-263, where we recommend lowering the parameter $\tau$ and restarting for the current round. To resolve your concern, we conducted experiments with the alternative implementation for $\tau=0.9$. We decreased $\tau$ by 0.05 when the constraints were not satisfiable. The average success probabilities were then 0.714 and 0.727, which do not exhibit a drastic loss of performance.
>
> > Q3. Is the current framework restricted to probabilities of belonging to a state?
>
> No, the framework can be easily extended to handle other objectives. The SRM modeling and optimizing for an objective are two relatively independent components. For example, we can replace the constraints on the probability of belonging to a state in Eq. (6) with constraints on other user-defined expected utilities. The following Bayesian updates and decision-finding can readily adapt to this scenario with some modifications. The reason we currently use the probabilities of belonging to a state instead of other expected utilities is that, in this initial work, we consider the "undesired future" as a discrete binary variable. More general extensions, including using other reward functions and different action costs, are reserved for future studies and exploration of the rehearsal learning framework.
>
> Thank you again for your insightful feedback! We hope that the explanations address your concern appropriately. We will also incorporate the above discussions into the revised version of the paper.

---

> > ### Comment · Reviewer_9i1V · 2023-08-19
> > **Thank you for the answers**
> >
> > Dear authors,
> >
> > Thanks a lot for the various answers, which confirms my positive opinion of the paper.
> >
> > I will not necessarily change my score, as I still think the paper deserves an accept, but I will raise my confidence level, as I am definitely more willing to defend my score and position thanks to the clarification.
> >
> > Best regards

---

### Official Review · Reviewer_LZtj · 2023-07-26

**Soundness:** 2 fair
**Presentation:** 2 fair
**Contribution:** 3 good
**Rating:** 5
**Confidence:** 3

**Summary:**

The paper argues that in decision-making, correlation is usually not enough but causation can be excessive. It introduces the idea of “rehearsation” which is a compromise between the two. Specifically, the paper proposes a novel rehearsal learning framework which models the interactions between interrelated (but not necessarily casually linked) variables in a dynamical system called structural rehearsal models (SRM). This frameowrk is applied to the problem of Avoiding Undesirable Future. A Bayesian inference framework is adopted to learn the graph posterior of the SRM. Mutual information maximisation is used as a criterion to select the alterations (similar notion to intervention in the causal setting) from a candidate set. Finally, a PAC bound is derived to quantify the decision risk in this framework.

**Strengths:**

* The paper introduces the new paradigm of “rehearsation” to decision making problems. To the best of my knowledge, this is a novel contribution.
* The PAC bound on the posterior is useful analysis and also practical in terms of quantifying uncertainty.
* Considering the linear case is a useful first step in building this framework.

**Weaknesses:**

* The main issue for me is the lack of comparison to similar methods in the causal inference literature, e.g. Causal Bandits/BO. The rehearsation framework is presented as a more sound framework for tacking AUF, but causal method can still be used as a practical for this problem even if they are philosophically not the principled methods to use. I would have liked to see a comparison against at least one prominent method in this area such as Causal BO.

* Similarly to my point above, I would have liked to see a bit more discussion on the strengths and limitations of this work compared to other methods that the authors mention such as Reinforcement learning and Causal Bandits. This could be on a practical level rather than a theoretical one.

**Questions:**

* How does the algorithm scale with the number of variables in the graph?
* Can the rehearsation formulation be used for other decision making problems than AUF?

**Limitations:**

Please see weaknesses section

---

> ### Author Rebuttal · Authors · 2023-08-06
>
> Thanks for your constructive comments! Below we address your questions in a point-by-point fashion.
>
> > W1. Regarding comparison to methods in causal inference literature, e.g. Causal Bandits/BO.
>
> Thanks for your suggestion! We need to clarify that causal bandits (CB) and causal Bayesian optimization (CBO) methods are not applicable to AUF for two reasons.
>
> Firstly, CBO and some CB methods assume known causal graphs [1,2,3]. However, the causal graph is not provided as input in AUF, and a causal graph that faithfully describes the relations among the variables may not exist at all (e.g., the interrelated relations captured by rehearsal graphs). As causal graphs are not available in AUF, such methods are not applicable. Secondly, the optimal decisions in AUF depend on the observed evidence $X$. However, current CB methods do not consider the existence of $X$ and only seek an optimal arm that maximizes the expected reward. Such approaches are unlikely to yield good results, as there is no universally optimal decision (arm) for all possible circumstances represented by $X$.
>
> To verify the above points, we conducted new experiments with a recent CB method that can handle unknown causal graphs [4]. As we had anticipated, the average success probabilities were respectively 0.073 and 0.066 for two datasets, which are far from satisfying, indicating that CB is not suitable for AUF.
>
> Additionally, we want to emphasize that the main purpose of the experiments was not to show superior performance but rather to demonstrate the feasibility of solving decision-making problems with rehearsal modeling. The proposed framework yielding favorable results indicates that this work could lay a foundation for promising future investigations into this topic.
>
> > W2. Regarding the strengths and limitations.
>
> The strengths of our work stem from the innovative use of rehearsal relations, which effectively help reduce the required number of interactions, as demonstrated in the experiments. As mentioned in Sec. 1, the success of reinforcement learning (RL) relies on a large number of interactions with the environment, which is feasible for game playing but can be unsuitable for problems like AUF, where interactions are sparse. In such cases, leveraging structural knowledge to enhance decision-making is a natural consideration. This aspect shares some similarities with causal bandits, where structural causal knowledge is used. However, causal knowledge can be too excessive and is difficult to obtain. Thus, we propose using the rehearsal relation, offering a more flexible approach yet still providing decision-making benefits. Another advantage of our proposed method is its ability to make decisions in the face of uncertainty, as it does not presume knowledge of the true underlying graph.
>
> As for limitations, a notable one of the current work, which could be addressed in future studies, is that sequential decision-making has not been considered, while RL methods excel in handling such scenarios. We think this limitation does not interfere with the main purpose of this work, which is to demonstrate the feasibility of rehearsal learning. We believe that this is a promising aspect that can be addressed in future research.
>
> > Q1. How does the algorithm scale with the number of variables in the graph?
>
> This is indeed an important question in practice. Since the exact characterization of the framework is intricate, we next provide a simplified analysis.
>
> Let $d$ denote the number of variables. We divide the running time into two parts. The first part is learning $\mathcal{G}$. For a practical graph learning procedure, it usually has a running time of $O(\text{poly}(d))$. Building $\mathcal{G}$ therefore consumes $O(|\mathcal{G}|\text{poly}(d))$ time. The second part is the update and decision finding in each round. Consider a simplified setting with only directional edges. The number of parameters is $O(d^2)$ as there are at most $O(d^2)$ edges. Let $O(B)$ denote the running time of a Bayesian updating method for one equation, like Bayesian linear regression. The time complexity of decision finding is $O(d n \log n)$ (see Sec. 4.2). The time complexity of the second part is then roughly $O(B d^2 + n' |\mathcal{G}| + d n \log n)$, where $n'$ is the number of samples for updating the graph posterior. The overall complexity for $T$ rounds is thus $O(|\mathcal{G}|\text{poly}(d)+T(B d^2 + n' |\mathcal{G}| + d n \log n))$, which scales polynomially with $d$. When considering bi-directional edges, it is reasonable to restrict the size of cliques to a constant to reduce modeling complexities. In this case, the number of parameters can be bounded by $O(poly(d))$, and the overall running time could still scale polynomially with $d$.
>
>
> > Q2. Can the rehearsation formulation be used for other decision-making problems than AUF?
>
> Yes, the formulation can adapt to various decision-making problems. The discussion primarily focuses on AUF because the scarcity of interactions in AUF necessitates the use of structural knowledge, and the rehearsation formulation is a good choice. But the rehearsation formulation, especially the SRM, is quite general. We can describe the variables in other decision problems with SRMs as well and modify the objective functions. For example, if the goal is to maximize the outcome instead of AUF, the expected outcome could be placed into the constraint Eq. (6), and the following optimization steps still apply.
>
> Thank you again for providing valuable feedback! We hope that the explanations address your concern appropriately. Some of the above discussion will also be incorporated into the revised version of the paper.
>
> [1] Aglietti et al. Causal Bayesian optimization. AISTATS, 2020.
>
> [2] Sussex et al. Model-based Causal Bayesian Optimization. ICLR, 2023.
>
> [3] Lee and Bareinboim. Structural Causal Bandits: Where to Intervene? NeurIPS, 2018.
>
> [4] Malek et al. Additive Causal Bandits with Unknown Graph. ICML. 2023.

---

### Decision · Program_Chairs · 2023-09-21

**Decision:**

Accept (poster)

**Comment:**

This paper proposes a new rehearsal learning framework to avoid the undesired future (AUF) in decision making that studies the relationship of variables between definitions of correlation and causality. It presents Structural Rehearsal Model (SRM) that can handle dynamic graphs in contrast to traditional Structural Causal Model (SCM) with static graphical models. This work provides theoretical analysis of the new framework and conducts preliminary experiments on synthetic data to showcases the feasibility of building decision-making models based on rehearsal relations.

All reviewers appreciates the novelty of this new problem setting and multiple reviewers increase their ratings after rebuttal. All reviewers agree on the acceptance of this work. Reviewer LgnQ has a remaining concern that the proposed framework is designed for the particular problem domain (AUF) that might be too small and impact its broader impact, but he/she does not consider it to be a criterion against the acceptance of this submission.